# LingOly-TOO: Disentangling Reasoning from Knowledge with Templatised Orthographic Obfuscation

**Jude Khouja**[1,†,*], **Lingyi Yang**[1,2,†], **Karolina Korgul**[1,†], **Simi Hellsten**[3,4,†], **Vlad Neacşu**[5,6],
**Harry Mayne**[1], **Ryan Othniel Kearns**[1], **Andrew Bean**[1,‡], **Adam Mahdi**[1,‡,*]

[1]University of Oxford,  [2]University of Nottingham,  [3]University of Glasgow
[4]United Kingdom Linguistics Olympiad,  [5]Asia-Pacific Linguistics Olympiad,
[6] 'Iorgu Iordan - Al. Rosetti' Institute of Linguistics of the Romanian Academy

🌐 oxrml.com/lingoly-too    ⭘ jkhouja/LingOly-TOO    🤗 jkhouja/lingoly-too

## Abstract

Frontier language models demonstrate increasing ability at solving reasoning problems, but their performance is often inflated by circumventing reasoning and instead relying on their expanding knowledge and memorisation capacity. We introduce LingOly-TOO, a challenging reasoning benchmark of 1,203 questions and a total of 6,995 sub-questions that counters these shortcuts by applying expert-designed obfuscations to Linguistics Olympiad problems. These obfuscations preserve the underlying solution logic while reducing the likelihood problems are solvable with via knowledge or memorisation. Our experiments show that models exploit shortcuts on the original question as performance markedly drop upon obfuscation. Even the best reasoning models remain highly sensitive, with scores dropping from around $0.59$ on original problems to $0.48$ after obfuscation. LingOly-TOO disentangles reasoning from knowledge, offering a clearer measure of true reasoning capabilities.

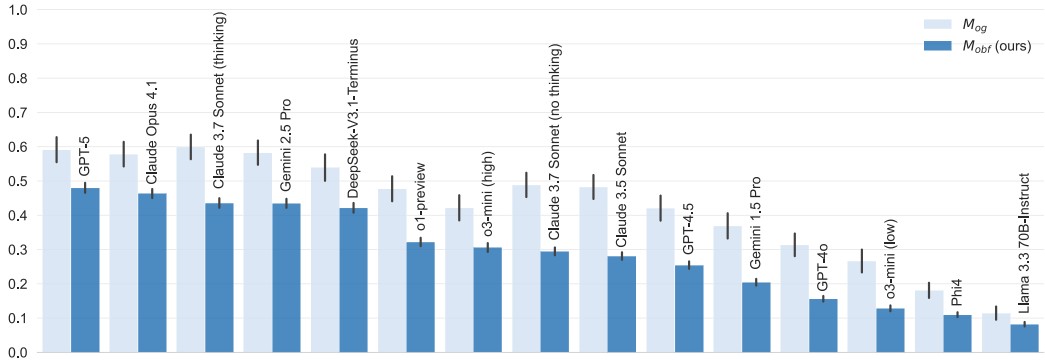

Figure 1: **Reasoning performance on LingOly-TOO.** Results without controlling for knowledge and memorisation abilities overestimate reasoning abilities (light blue). Obfuscation mitigates this effect and offers improved estimates (dark blue). There are two scores per model: $M_{og}$ is based on the original problems and $M_{obf}$ is based on obfuscated problems.

## 1 Introduction

Experts often define reasoning as the ability to apply abstract rules to derive novel judgments (Koralus, 2022; Huang & Chang, 2023; Lampinen et al., 2024). This ability includes inductive, spatial, causal,

---

[†]Equal first author contribution

[‡]Equal last author contribution

∗ Correspondence to: <jude.khouja@oii.ox.ac.uk> or <adam.mahdi@oii.ox.ac.uk>

and commonsense reasoning skills (Kazemi et al., 2025). Benchmarks aim to measure reasoning by testing models on unseen tasks. For such measurements to be valid, reasoning must be both a necessary and sufficient condition for success. However, as training sets grow in size and models' memorisation capacity increases, distinctions between train and test sets, and in- and out-of-domain tasks are blurred (Raji et al., 2021), leading to bias in benchmark estimates.

We distinguish two capabilities resulting from training LLMs that help models solve linguistic tasks. We refer to *knowledge* as information stored in model parameters which captures linguistic, factual, and commonsense patterns useful for downstream tasks. Stored knowledge, while generally useful, can confound the measurement of symbolic reasoning capabilities. For example, GPT-5 can already translate from Welsh to English, hence a linguistic reasoning task based on deriving rules from paired Welsh–English translations would be trivial. We refer to *memorisation* as when models exploit prior exposure to evaluation datasets and report answers based on previously seen problems in training (Li & Flanigan, 2024; Zhou et al., 2023).

Consequently, and amidst the rapid saturation in reasoning benchmarks (Kazemi et al., 2025; Phan et al., 2025), novel methods for generating challenging, unbiased benchmarks are paramount for measuring true reasoning abilities. To prevent models from bypassing reasoning, test cases should minimise the chance that problems are answerable using knowledge or memorisation. Consider the following scenario: a user asks a large language model (LLM) to solve a language puzzle, but it is written in an unfamiliar script or orthography.[1] In this situation, the model's knowledge of the language is rendered useless. The model must deduce grammatical rules and patterns from context, then apply them to solve the given puzzle. Prior work has explored using synthetic test cases (Saparov & He, 2023) to evaluate reasoning. Similar interventions have also been applied to logic puzzles and mathematics, e.g. using symbolic templates to permute variable values (Mirzadeh et al., 2024). However, these small-scale perturbations are less effective because questions remain similar to examples the model may have previously encountered. By contrast, permuting the orthography of a language results in test cases with minimal chance of appearing in training corpus, while the underlying solution logic remains intact.

Our benchmark LINGOLY-TOO operationalises this idea. Original problems are taken from the UK Linguistics Olympiad (UKLO (United Kingdom Linguistics Olympiad, 2023)), which can be solved by high-school students without specific linguistic or domain knowledge. The problems undergo obfuscation: linguistically-informed grapheme-level permutations that preserve the underlying solution logic, remove clues a model could match to prior knowledge, and provide thousands of distinct, solvable variants. Our benchmark aims to test whether a model can use symbolic inductive reasoning to infer the abstract rules and patterns needed to answer the questions from context.

Our evaluation results show that models often rely on language knowledge rather than reasoning, inflating apparent ability, especially in high-resource languages. Although reasoning models, trained to leverage inference-time compute, outperform general-purpose LLMs, they remain sensitive to permutations and exhibit failures such as inconsistent reasoning loops. LINGOLY-TOO addresses key issues in benchmarking LLMs: saturation, contamination, and weak construct validity (Reuel-Lamparth et al., 2024). Our primary contributions are as follows:

- **An unsaturated benchmark for frontier reasoning models.** The top language model on our benchmark, GPT-5, scores 48% overall and only 31% on the highest difficulty problems (Section 4.1).

- **A method to quantify knowledge effects.** The difference between performance on the unperturbed and perturbed problems highlights reasoning shortcuts. We show score inflation from knowledge is correlated with language resourcedness (Section 4.2), and that providing the correct reasoning logic mitigates it (Section 4.3).

- **A method for generating new uncontaminated reasoning problems.** Experiments with then-unpublished UKLO 2025 problems show that perturbation-related performance drops persist, indicating that the effects are not due solely to training-set overlap (Section 4.4).

---

[1]System for writing a language. A glossary of linguistic terms used in this article is included in Appendix A.

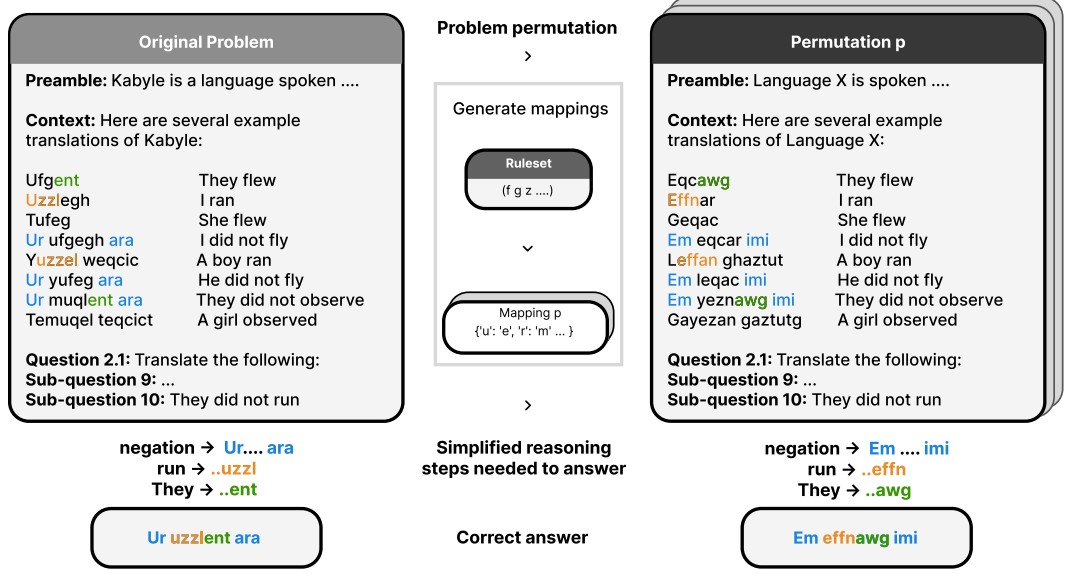

Figure 2: **Obfuscation example.** Problem 160 (Kabyle) before (left) and after (right) obfuscation with the simplified inductive reasoning steps needed for answering sub-question 10. For each obfuscation, we sample a character mapping (permutation) based on the ruleset (see Appendix B.3 to learn more about rulesets), then apply it to obfuscate the problem and the answer. Only the original problem can be solved with the aid of model's internalised knowledge.

## 2 RELATED WORK

### 2.1 REASONING IN LLMS

LLMs show rapid progress on task-reasoning across domains, such as mathematical word problems (Cobbe et al., 2021; Hendrycks et al., 2021; Chen et al., 2023; Shao et al., 2024), visual pattern-matching (Chollet, 2019), multi-step planning (Valmeekam et al., 2022; Kambhampati, 2024), and commonsense question-answering (Zelikman et al., 2022; Lin et al., 2020; Rajani et al., 2019; Talmor et al., 2019). It remains unclear to what degree these abilities generalise to novel domains and tasks. Models fail in systematic ways, showing distractability (Shi et al., 2023), content effects (Lampinen et al., 2024), and sharp drops under minor perturbations (Mirzadeh et al., 2024). They also rely on superficial patterns (token bias) (Jiang et al., 2024), and this reliance persists despite inference-time scaling (McCoy et al., 2024). This is further complicated by dataset contamination, where benchmark questions appear in training data (Zhou et al., 2023; Jacovi et al., 2023; Yang et al., 2023). The evidence suggests that high LLM benchmark scores are inflated through shortcuts (Bean et al., 2024).

### 2.2 LINGUISTIC REASONING WITH LLMS

Linguistic reasoning problems test the in-context learning and reasoning ability of LLMs by providing examples of a language and tasks to accomplish based on linguistic analysis (Bean et al., 2024; Tanzer et al., 2024). These problems require a mix of deductive, abductive, and analogical reasoning, allowing broad relevance beyond linguistics (Bean et al., 2024). Linguistics Olympiads are a common source of these problems, where puzzles have been curated for student competitions, ensuring the quality of the puzzles as well as their solvability by someone with no prior knowledge of the languages in question (Şahin et al., 2020; Chi et al., 2024). Morphological complexity has been shown to be particularly challenging for LLMs, as well as languages with lower similarity to English (Choudhary et al., 2025). Advanced problems from these competitions typically draw on low-resource languages where LLMs have minimal pre-training exposure, and remain difficult for top models (Sánchez et al., 2024; Lian et al., 2025). However, models show signs of partial contamination even in low resource languages (Bean et al., 2024), indicating a need for further methods to improve benchmark validity.

# 3 LINGOLY-TOO BENCHMARK

LINGOLY-TOO is a reasoning benchmark extending the LINGOLY benchmark (Bean et al., 2024). It consists of 1,203 questions and a total of 6,995 sub-question and answer pairs, generated through an obfuscation process designed by experts to preserve the intrinsic reasoning steps needed to solve each question, and to generate novel test data that measure the generalisable reasoning abilities of models.

## 3.1 LINGUISTICS TASKS

The benchmark is adapted from 82 problems from the UKLO. Linguistics Olympiad problems have several desirable features. All problems are grounded in natural languages, but require no preliminary knowledge in linguistics or any other domain beyond what is expected from middle and high school students. Each problem is self-contained and can be solved from the provided context using general reasoning and pattern matching skills. Answers vary in format, including free response and multiple choice, and can be evaluated with simple rule-based matching. Problems span five difficulty levels (Breakthrough, Foundation, Intermediate, Advanced, Round 2) sourced from UKLO.[2]

A typical problem sheet (referred to hereafter as a *problem*) consists of multiple questions, each question consists of multiple *(sub-question, answer)* pairs. Each problem presents words or phrases in a language of focus (termed *Problemese*) alongside their translations into the language in which the problem is written (termed *Solverese*) (Bozhanov & Derzhanski, 2013). The solver should deduce the underlying patterns and structures and use them to produce a best answer for each sub-question in a manner that resembles few-shot learning (Figure 2).

## 3.2 REASONING-EQUIVARIANT PERMUTATION

Although the original 82 problems include low-resource languages, these languages are increasingly represented in pre-training datasets, allowing models to rely on memorisation or knowledge rather than reasoning. We address this challenge by generating *reasoning-equivariant permutations* that obfuscate the original languages while preserving the underlying language mechanisms and solution steps. In Linguistics Olympiads, *obfuscation* describes the process that prevents cheating by removing metadata such as language names, language families, and geographic information that could identify the Problemese language (Asia Pacific Linguistics Olympiad, 2024; United Kingdom Linguistics Olympiad, 2021). We extend this approach by obfuscating the Problemese text itself, making it unlikely that models can solve problems using knowledge or memorisation.

Because the problems require symbolic reasoning over subwords (both morphemes such as suffixes, and phonemes), common permutation techniques that operate on complete words such as synonym replacement, paraphrasing, or word swapping are not suitable. These would corrupt the symbolic units and invalidate the problems. Instead, we manually created a ruleset for each problem to generate valid permutations of the targeted tokens (Problemese text). Valid permutations treat graphemes (both single letters and letter-combinations like English *th* and *sh*) as single units, and preserve any relationships between them which are relevant to solving the problem. They also preserve any loanwords or English cognates that might aid in solving the problem. Names of people, deities, and sacred places were also kept unchanged.

It is important to note that naive permutation of graphemes can easily render the problem unsolvable if it does not preserve the properties used in the underlying linguistic mechanism. We illustrate this with the example of minor vowel harmony highlighted in Problem 5 (Turkish, by Bozhidar Bozhanov). The suffix *-sVz* (meaning 'lacking', similar to the English '-less') can take four different forms depending on the previous vowel (Table 1). From a linguistic perspective, each of the four vowel pairs (*e,i*), (*o,u*), (*ö,ü*), (*a,ı*)

| Last vowel | Suffix |
|---|---|
| *e* or *i* | *-siz* |
| *o* or *u* | *-suz* |
| *ö* or *ü* | *-süz* |
| *a* or *ı* | *-sız* |

Table 1: **Problem 5 (Turkish) suffix form.** Suffix form of *-sVz* changes depending on the last vowel.

---

[2]For problems marked with multiple levels, we take the lower one. The distribution statistics of these difficulty levels can be found in Appendix D.

differ in the shape of the lips (*roundedness*) and position of the tongue (*backness*) when that sound is produced. For example, (*e*,*i*) are [−ROUND] and [−BACK], while (*o*,*u*) are [+ROUND] and [+BACK]. Any permutation which did not preserve these pairs would render the problem much harder and possibly unsolvable, since the vowel in the suffix would not correspond as naturally to the previous vowel. As such, all possible permutations must preserve the pairings, although the pairs may be permuted amongst themselves.

Our expert list of valid permutations takes into account the interactions between different segments to maintain two principles: (i) ensuring the permutations are reasoning-equivariant, and (ii) obtaining the maximum number of possible permutations. Further details can be found in Appendix B.

### 3.3 DATA GENERATION

We manually annotated 1,005 (sub-question, answer) pairs from 82 problems[3] with special tags for later processing. Metadata such as language names, language families, and geographic information, which had no impact on problem solvability but may provide clues for LLMs to use shortcuts were removed. A detailed description of the annotation process, including examples, is provided in Appendix C.

We randomly sampled up to 6 valid permutations per problem, and generated obfuscated versions of the problem by altering the Problemese text according to the selected permutations. Finally, we removed any special characters used for annotations. The original questions and the 6 permutations resulted in 6,995 (sub-question, answer) pairs across all problems.[4]

### 3.4 EVALUATION

**Metrics** We define a correct prediction as an exact match to the answer, with no credit given to partially correct answers. The correct answer often involves making small changes to words already present in the problem, so scores that offer partial credit may reward incorrect answers resulting from wrong reasoning.

Let $L(i, j, k, p)$ be the exact match score for problem $i \in \{1, ..., 82\}$, question $j \in \{1, \ldots, n_i\}$, sub-question $k \in \{1, \ldots, m_{i,j}\}$ after applying permutation $p \in \{0, \ldots, 6\}$, where $n_i$ is the total number of questions in problem $i$ and $m_{i,j}$ is the total number of sub-questions in problem $i$ question $j$. To simplify notation, permutation $p = 0$ is the identity, i.e. the unpermuted case. We define $M_{obf}^{i,j}$ to be the average exact match score across all sub-questions in all (non-identity) permutations of problem $i$ question $j$, that is, $M_{obf}^{i,j} = (1/(6m_{i,j})) \sum_{k=1}^{m_{i,j}} \sum_{p=1}^{6} L(i, j, k, p)$. In analyses where we compare results to those on the original (unpermuted) problems, we define $M_{og}^{i,j} = (1/m_{i,j}) \sum_{k=1}^{m_{i,j}} L(i, j, k, 0)$ to be the average exact match across all sub-questions in the unpermuted problem $i$ question $j$. To evaluate a single model on the entire benchmark, we report the average across all problems and questions, that is

$$M_{obf} = \frac{1}{82} \sum_{i=1}^{82} \frac{1}{n_i} \sum_{j=1}^{n_i} M_{obf}^{i,j} \qquad \text{and} \qquad M_{og} = \frac{1}{82} \sum_{i=1}^{82} \frac{1}{n_i} \sum_{j=1}^{n_i} M_{og}^{i,j}. \qquad (1)$$

**Evaluation pipeline** We broke down problems into individual questions and prompted models on each question with all of its (sub-question, answer) pairs. Each prompt consisted of a *preamble* (detailing background information about the language), *context* (key information the models need to reason over and infer the correct rules), all questions in the problem (some questions depend on previous ones) and the specific question to be answered. The prompt also specified the expected `json` structure response. The exact template used is provided in Appendix E. Since our aim is reasoning rather than instruction-following, we applied rule-based post-processing to recover improperly formatted responses. Remaining unparsed responses were counted as incorrect and detailed in Appendix G.

---

[3]We refer to problems by their numbering on the UKLO past problems database (United Kingdom Linguistics Olympiad, 2023).

[4]This is less than $7 \times 1,005 = 7,035$ as the total possible permutations for some problems are less than 6.

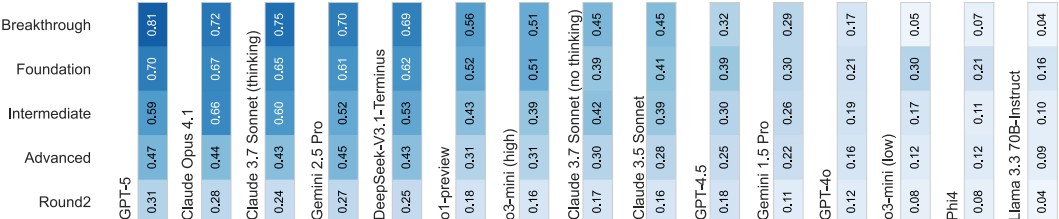

Figure 3: **Breakdown of $M_{obf}$ scores by difficulty level.** For the stronger models, we see a clear decreasing trend in scores with increasing problem difficulty.

**Models** We evaluated fifteen reasoning and general-purpose model variations including open-source and closed-source models on the benchmark. See Appendix F for further details about the models.

### 3.5 BENCHMARK VALIDATION

Several measures were taken to ensure the quality and correctness of the benchmark. All problem annotations were performed manually and subsequently validated by two team members with expertise in Linguistics Olympiads. In addition, automated checks were incorporated to detect issues. Any identified errors triggered re-annotation and validation. A second round of both manual expert review and automated validation was conducted on the final data.

Obfuscation was designed to preserve reasoning steps and verified by team members with Linguistics Olympiads expertise. Additionally, two International Linguistics Olympiad (IOL) medallists, familiar with the original UKLO problems, audited a sample of obfuscations and independently confirmed that all problems remained solvable via the same reasoning steps.

Because obfuscated text no longer adhered to the original languages, it could impose a cognitive penalty in humans who may be more familiar with the original language orthographies. To estimate this effect, we ran a randomised controlled trial on a sample of six problems (chosen to be relatively easy to ensure novices had a realistic chance of solving them) with 172 human subjects. Participants were randomly assigned to either an original or obfuscated problem. Performance decreased $5.70\%$ on obfuscated problems, likely due to their unfamiliarity, as noted by one auditor. Details of the study and analysis are provided in Appendix L.

## 4 EXPERIMENTS AND ANALYSIS

### 4.1 OVERALL PERFORMANCE

Figure 1 shows all models' results on LINGOLY-TOO. The $M_{og}$ metric (no obfuscation), is reported for comparison, which we posit is an over-estimation of reasoning abilities. We see that frontier models achieve $\sim 0.59$ on $M_{og}$. However, using $M_{obf}$ metric (with obfuscation) , this drops to a maximum of 0.48. $M_{obf}$ scores also reflect improved reasoning capabilities of reasoning models over general-purpose models. For example, GPT-5 scores higher than GPT-4.5, and Claude 3.7 (thinking) outperforms Claude 3.7 (no thinking) (0.44 vs. 0.30). In addition, o3-mini (high) outperforms o3-mini (low) by a larger margin (0.31 vs 0.13), indicating that increasing inference-time reasoning budget is useful for the benchmark tasks. Performance also seems related to the difficulty level in the best performing models but the pattern is not consistent in the weaker models (Figure 3). GPT-5 and Claude 3.7 Sonnet score highly (0.81 and 0.75, respectively) on *Breakthrough* problems but score below 0.31 at the *Round 2* level, demonstrating that the benchmark is far from saturated and offers a range of difficulties valuable for tracking progress. Overall, no model scores over 0.5, suggesting that despite recent breakthroughs, multi-hop inductive reasoning remain an open challenge.

| Answer (% of total) | GPT-4o | | Llama-3.3 70B | |
|---|---|---|---|---|
| | $M_{og}$ (SE) | $M_{obf}$ (SE) | $M_{og}$ (SE) | $M_{obf}$ (SE) |
| All | 0.08 (0.00) | 0.02 (0.00) | 0.05 (0.00) | 0.03 (0.00) |
| Digit (13.9%) | 0.16 (0.01) | 0.08 (0.00) | 0.19 (0.01) | 0.11 (0.01) |
| Single Char (14%) | 0.03 (0.00) | 0.03 (0.00) | 0.05 (0.00) | 0.04 (0.00) |
| Y/N (0.6%) | 0.62 (0.04) | 0.31 (0.04) | 0.19 (0.04) | 0.49 (0.04) |
| Other (71.6%) | 0.07 (0.00) | 0.01 (0.00) | 0.02 (0.00) | 0.01 (0.00) |

Table 2: **Performance in the *no context* setting.** In the *no context* setting, prompts contain insufficient information to answer questions. Scores are categorised by answer type.

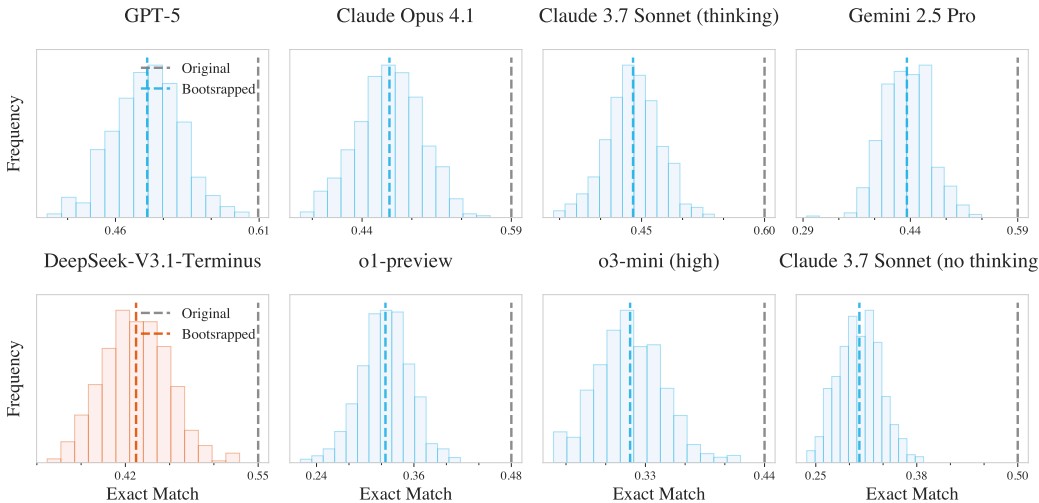

Figure 4: **Score distribution across bootstrapped samples.** Distribution of scores across 500 bootstrapped samples of our data by model. Each consists of 82 problems. Open source models are shown in orange while proprietary models are in blue (for full results, see Appendix I).

## 4.2 KNOWLEDGE VS. REASONING ABILITIES

**Ability to score without reasoning** A key goal of the benchmark is to minimise the effects of models' knowledge on reasoning estimates. We evaluate LLMs' ability to score without reasoning on obfuscated data by adopting the *no context* setting from Bean et al. (2024), where critical information is removed from context, rendering the problems unsolvable by reasoning alone. We conduct a small experiment using 30 permutations per problem with two models (Llama 3.3 70B and GPT-4o).

Table 2 shows the results in the *no context* setting. $M_{obf}$ for Llama 3.3 70B and GPT-4o is 0.02 and 0.03 respectively. When only considering questions with lower chance of random guessing (Other), $M_{obf}$ further drops to $\sim 0.01$ in both models, demonstrating the effectiveness of obfuscation. This is not the case in the unobfuscated questions where GPT-4o is more successful in circumventing reasoning, scoring $M_{og} \sim 0.07$ through knowledge and memorisation. Inspecting a few examples where GPT-4o was successful reveals cases where GPT-4o correctly answers through direct translation, such as generating the correct Welsh translation: *Aeth meddyg i Gymru* to the English sentence *A doctor went to Wales*.

**Reasoning gap** To compare benchmark estimates with those from only the unobfuscated questions, we construct a bootstrap-style distribution over permutations. Specifically, we create 500 bootstrap sets, each containing all 82 problems. For each problem, we independently and uniformly select one of its seven versions (original plus six permutations) and compute the score. Figure 4 shows the distribution of scores on bootstrapped sets. When using unobfuscated problems, the score ($M_{og}$) consistently appears at the right tail of the distribution, reflecting a significant decrease in models'

| Score | Standard | Dash | Character |
|---|---|---|---|
| Original | 0.087 | 0.051 | 0.053 |
| Permuted | 0.050 | 0.045 | 0.035 |

Table 3: **Exact match score with varying tokenisation.** 'Dash' tokenisation inserts a dash between each character in the Problemese data. 'Character' tokenisation forces the model to tokenise each character individually.

performance with obfuscation. A similar gap is observed in the subset of problems used in the human study, where average model performance falls from $0.45$ to $0.37$ ($8.6\%$) on obfuscated problems. However, reasoning models incur a smaller performance drop, on par with humans ($5.8\%$), suggesting that reasoning models have reduced the gap in solving problems in their original orthography versus obfuscated versions compared with general-purpose models.

**Tokenisation effect**  A possible explanation of the performance drop is that uncommon character sequences would affect LLMs through tokenisation by producing unuseful token representations. As tokenisers of LLMs are trained using frequency statistics on languages in their original orthography, sub-optimal tokenisation could explain the fall in model performance. We conduct a small experiment on a multilingual model (Aya-23-35B) to compare three variations of tokenisation: standard tokenisation, introducing a dash between each character in the Problemese and tokenising each character separately. Table 3 shows that enforcing separation of the tokens does not improve model performance. For both the original and obfuscated versions, exact match score does not improve if the tokenisation is altered, pointing towards explanations of failed reasoning other than the standard tokenisation. Additional tokenisation results are detailed in Appendix K.

**Variance across permutations**  Models score higher on unobfuscated problems, possibly by utilising their knowledge from exposure to the original languages. Since obfuscation preserves solution logic, we quantify the performance difference due to obfuscation. For a fixed permutation $p$ of problem $i$, we define $\Delta_{obf}^{i,p}$ as the difference between the average score across all questions of problem $i$ under obfuscation $p$ and the average score of all questions for the unobfuscated problem. That is

$$M_{og}^i = 1/n_i \sum_{j=1}^{n_i} M_{og}^{i,j}, \quad \text{and} \quad \Delta_{obf}^{i,p} = \frac{1}{n_i} \sum_{j=1}^{n_i} \left( \frac{1}{m_{i,j}} \sum_{k=1}^{m_{i,j}} L(i,j,k,p) \right) - M_{og}^i.$$

Then $\Delta_{obf}^{i,p}$ is negative when the model performs worse on a problem permutation, zero when there is no difference, and positive otherwise. Careful inspection reveals that although on average these differences are negative, model performance gaps vary across problems as well as across permutations within each individual problem. In a few cases such as Problem 24 (Tadaksahak) and Problem 67 (Navajo), some permutations even led to improved performance. We also see problems that had similar effects upon permutation across several models, indicating similarities in model failure modes. Appendix H includes full analysis of $\Delta_{obf}^{i,p}$ across all generated permutations of the 82 problems.

**Language resourcedness effect**  Problems from higher resource languages generally show large negative $\Delta_{obf}^i = 1/6 \sum_p \Delta_{obf}^{i,p}$ values across all models such as Problem 74 (Japanese): $-0.59$, Problem 76 (Finnish): $-0.59$ and Problem 178 (Italian): $-0.57$. We approximate language resourcedness using the number of speakers of the language (in logarithmic scale). To quantify its effect on the gap in reasoning estimates, we apply linear regression to predict $\Delta_{obf}^i$ from resourcedness, separately for each problem difficulty and for models grouped into reasoning and general-purpose categories. We apply the Bonferroni correction to account for multiple-testing when reporting significance.

Results (Figure 5) show that language resourcedness is negatively correlated with reasoning performance gaps at the higher difficulty levels. In reasoning and general-purpose models, coefficients are negative ($\beta < 0$, $p < 0.01$ for Advanced and Round 2 problems). One possible explanation is that when models are unable to reason at higher difficulty levels, models are better at guessing the answer in languages with larger amount of data available for training.

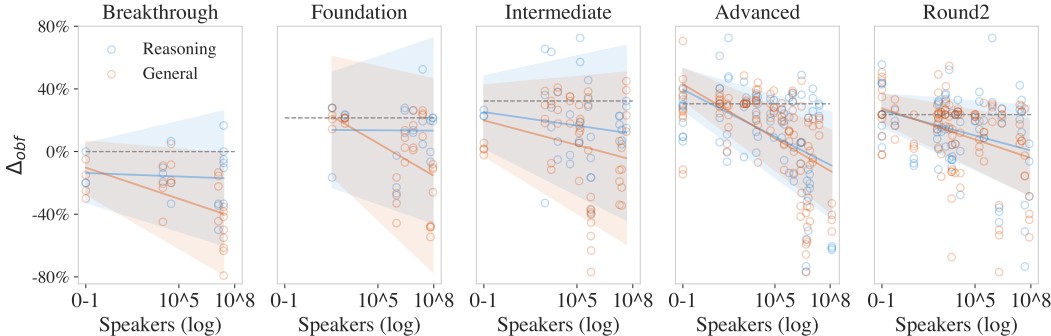

Figure 5: **The effect of permutation is larger for high-resource languages.** Each point represents all problems associated with a language of specific numbers of speaker. Solid lines are fitted regressions and shaded areas are the $95\%$ confidence intervals.

### 4.3 EVIDENCE OF NASCENT REASONING

Let us define

$$M_{rob}^i = \frac{1}{n_i} \sum_{j=1}^{n_i} \frac{1}{m_{i,j}} \min_p \sum_{k=1}^{m_{i,j}} L(i,j,k,p) \qquad M_{rob} = \frac{1}{82} \sum_{i=1}^{82} M_{rob}^i,$$

as a more robust metric of average model performance across all problems since it measures the minimum score achieved on any problem permutation. Under this metric, performance drops in all models (see Figure 6 for more details). The best model, GPT-5, only achieves $0.29$ (a decrease of over $0.19$). Reasoning models, which score the highest, are more consistent than general purpose models. An attempt for systematic analysis of reasoning traces using LLMs was unsuccessful as LLMs were unable to identify erroneous reasoning steps, making both type I and type II errors. We manually inspected the reasoning traces of Claude 3.7 (thinking) on two randomly selected problems where the model correctly answered questions in all permutations. We found examples where the model can identify subwords and suffixes across all permutations, infer their meaning, and deduce the correct answers (e.g. excerpt from one trace: *The verb prefixes seem to agree with the subject: "si-" for "they"*). However, these positive examples of a generalisable reasoning ability were among many more failed cases. Among them are traces with repeated analyses, sometimes drawing the same conclusions, others drawing different ones, even within the same permutation. More generally, $M_{rob}^i$ shows that all models perform noticeably poorly, underscoring that reasoning is brittle and remains an open challenge to frontier models.

**Reasoning with expert guidance**    By prompting with guidance steps written by team members based on the original problems, we evaluated whether model performance on the permuted problems improved. Using two problems chosen to have concise solutions (Problems 69 and 164) and four models from several providers (Claude 3.7 (thinking), Claude 3.5, Gemini 1.5 Pro, o3-mini high), we find that all models improve with the mean score on permuted questions ($M_{obf}$) increasing ($0.66$ to $0.76$). This experiment suggests that, as in domains such as mathematics, multi-hop reasoning on unfamiliar language problems benefits from improved intermediate inductive reasoning steps.

### 4.4 KNOWLEDGE VS MEMORISATION

While we assume a possibility that models are exposed to some of the unpermuted problems during training, we do not think that this accounts for total performance gaps. To illustrate, we were able to access and evaluate a subset of models on 5 problems from the UKLO 2025 that had not yet been published online at the time of the experiment and are of a higher difficulty level. Results in Table 9 show a comparable performance drop between the unpermuted and permuted cases to that observed in the benchmark. The difference in scores illustrates that the performance decrease is not simply due to memorising answers since these problems have not been included in models training. The gap also highlights the effectiveness of our permutation to control for knowledge even on unseen problems.

## 5 LIMITATIONS

Exact Match uniformly penalises all incorrect answers, including partially correct ones, and limits insights into failure modes. Yet, metrics such as BLEU (Papineni et al., 2002) and ROUGE (Lin, 2004) are unsuitable for the very short answers common in our benchmark (note from Table 2 that 14% of our answers are single digits) and chrF (Popović, 2015) is sensitive to repeating related words from the context, which is a common behaviour in the models being tested. For example, Problem 16 4.1 (3) requires a translation of "your (plural) iguana" to Ulwa. The word for iguana in Ulwa is kahma, which is already given in the context. The correct answer is kah**mana**ma, as your (plural) is -mana-. Claude Opus 4.1 returned kahma**na**, which is the wrong form and placement of the possessive. If we assigned partial credit through e.g. edit distance, the answer from Claude will score highly, even though this is an incorrect answer based on incorrect reasoning. Assigning partial credit would unnecessarily inflate the baseline through repeating parts of words in the context. Future work could explore alternative evaluation metrics that capture partial solutions or help discern partially correct answers to ensure construct validity (Bean et al., 2025). Novel evaluation scores could credit correct intermediate reasoning steps alongside the final answers.

While problems in LINGOLY-TOO do not depend on any specific domain knowledge, we limit this work to inductive and deductive reasoning in the natural language modality. The benchmark is not a comprehensive evaluation of all reasoning abilities across all modalities but results suggest that improvements in certain reasoning abilities would translate to improvements in others.

## 6 CONCLUSION

We introduced LINGOLY-TOO, a benchmark for evaluating reasoning in language tasks. By applying expert-designed obfuscations to Linguistics Olympiad problems, we generated permutations that preserved solution logic while remaining novel to models. Our results showed that the benchmark effectively controls for model knowledge and memorisation, revealing that LLMs (including reasoning models) often rely on shortcuts rather than genuine reasoning. We further quantified how language resourcedness amplify these shortcuts. Finally, we observed that reasoning models were better able to apply symbolic and inductive reasoning and that progress reported in domains such as mathematics and coding appears to partially translate to language problems.

Our benchmark remains unsaturated, especially at the highest problem difficulty, leaving ample headroom. Our results support existing claims of model reliance on shortcuts, which biases reasoning ability estimates. Compared to the unobfuscated problems, deriving estimates from obfuscated problems paints a more conservative picture of the reasoning ability of frontier models. We find that reasoning consistency and robustness remain limited in frontier models and warrant further attention.

### ETHICS STATEMENT

This paper aims to advance the field of machine learning. Constructing and publishing problems using low-resource languages to create a benchmark may lead to concerns about the exploitation of or other harm caused to the communities who use and own these languages (Tanzer et al., 2024). The problems in this benchmark are created either by native speakers or using published resources such as sketch grammars. In these cases, the relevant language communities have already given consent for a linguist to publish information on the language. In this paper, we transform existing puzzles, and do not create new content in the languages.

The obfuscations of the problem amount to a new orthography for the language, and do not change any other aspect of the language such as grammar or the meanings of words; do not alter the names of people, deities, or sacred places; and are not publicly available. All participants in the human study were provided with information about the language they had studied and its speakers after completing the study.

### REPRODUCIBILITY STATEMENT

We release complete anonymised code and evaluation scripts for our benchmark for reproducibility at https://github.com/jkhouja/LingOly-TOO.

ACKNOWLEDGMENTS

The authors would like to thank Riley Kong and Deeraj Pothapragada for their help with the external audit of the obfuscated problems. L.Y. was supported by EPSRC [EP/S026347/1] and the Hong Kong Innovation and Technology Commission (InnoHK Project CIMDA). H.M. is supported by ESRC [ES/P000649/1] and would like to acknowledge the London Initiative for Safe AI. A.M.B. and K.K. are supported by the Clarendon Fund Scholarships at the University of Oxford. A.M.B., K.K. and A.M. were partially supported by the Oxford Internet Institute's Research Programme funded by the Dieter Schwarz Stiftung gGmbH. R.O.K. is supported by the Clarendon Fund Scholarships and the Jesus College Old Members' Scholarship at the University of Oxford.

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

## A  GLOSSARY

| Word | Meaning |
|---|---|
| *cognate* | Word or word-part with the same historical origin as another. E.g. English *father*, German *Vater*, Swedish *far*. |
| *diacritic* | Symbol added to a letter or other basic glyph; often called 'accent'. |
| *(language) family* | Group of languages all derived from a common ancestor. E.g. the Romance language family is derived from Latin. |
| *grapheme* | Letter or group of letters that typically represent a single sound or suprasegmental feature. See Kohrt (1986) for a discussion of other definitions in common usage. |
| *lexeme* | Basic unit of the lexicon (vocabulary) of a language. E.g. *sit*, *sits*, *sitting*, *sat* are all inflected forms of the same lexeme. |
| *loanword* | Word borrowed from another language. E.g. Nepali *pēnsil* 'pencil'. |
| *morpheme* | Basic grammatical unit of meaning within a word. E.g. *help-less-ness* consists of three morphemes. |
| *morphology* | Branch of linguistics dealing with the internal structure and formation of words. |
| *orthography* | System for writing a language, including the choice letters or other glyphs and spelling conventions. |
| *phonetics* | Branch of linguistics dealing with the production and perception of speech sounds or signs. |
| *phonology* | Branch of linguistics dealing with the systematisation and organisation of sounds or signs within a language. |
| *phonological distinction* | Distinction between two speech sounds that is treated as meaningful within a language. |
| *Problemese* | Unknown language that is the focus of a Linguistics Olympiad problem. |
| *semantics* | Branch of linguistics dealing with the study of meaning. |
| *Solverese* | Language that a Linguistics Olympiad problem is written in; assumed to be the working language of the solver. |
| *suprasegmental* | Phonetic or phonological feature that extends beyond a single speech sound. E.g. stress, intonation, pitch. |
| *syllabification* | Procedure for forming valid syllables out of a string of speech sounds. |
| *syntax* | Branch of linguistics dealing with the organisation of words into phrases, clauses and sentences. |

Table 4: **Glossary of linguistic terms.**

## B  PERMUTATION RULESETS

The core of the obfuscation procedure is the altering of the Problemese data. This is done by creating a new orthography[5] for the Problemese language, so that the underlying grammar remains the same, but the data looks unlike any other existing data in the language. It is therefore impossible to solve the problem by relying on prior knowledge instead of reasoning. However, since this "re-spelling" preserves the important structure of the problem data, it preserves the solution steps.

### B.1  OVERVIEW

To create a new orthography, each grapheme[6] in the original orthography must be replaced with a new grapheme. We choose to use the same graphemes in the original and obfuscated orthographies, so that each new obfuscated orthography is simply a permutation of graphemes in the original orthography. We represent these permutations as mapping dictionaries from original graphemes to the new graphemes.

There are two possible effects of such an obfuscation that could increase the difficulty of a problem or even render it unsolvable:

(i)  Some problems rely on solvers recognising English cognates or loanwords, either explicitly or implicitly. We give two examples.

- Swedish (Problem 176, by David Hellsten) includes the following question. "One of these adjectives behaves slightly differently to the others. Which one? [...] It is closely related to an English adjective with the same meaning. Which one?" This relies on noticing that the Swedish adjective *lite-, lill-* (translated as 'small') is cognate to the English word *little*.
- Taa (Problem 217, by Ethan Chi) does not strictly require the solver to notice any cognates or loanwords; however, when the problem featured in UKLO, many solvers broke into the problem by observing that *bòkòsè* was likely a loanword that meant 'box'.

In both cases, an arbitrary new orthography would make it impossible to recognise the cognate or loanword. This would render Swedish impossible and Taa significantly more difficult.

(ii)  Many problems rely on solvers either deducing or already understanding the linguistic relationships between certain sounds. We gave one such example in Section 3.2, and give two more here, which are representative of the sorts of issues that can arise.

- In Tadaksahak (Problem 24, by Bozhidar Bozhanov), a specific form of a verb is formed by adding one of $\{s,z,\check{s},\mathbf{3}\}$ to the end of the word; solvers are told that $\check{s}$ is pronounced like 'sh' in 'shoe', and $\mathbf{3}$ like 's' in 'casual'. This is a plausible observation to make, since these are the four 'sibilant' (hissing) consonants, a fact which is usually taught in high school English classes, if not earlier.
- In Dinka (Problem 189, by Simi Hellsten), the solver must understand how vowels change in Dinka nouns and verbs to indicate different "grades" – singular and plural nouns, and different subjects for verbs. They must observe that vowels do not change randomly, but only raise or lower in the mouth:

$$i \leftrightarrow e \leftrightarrow \varepsilon \leftrightarrow a \quad \text{and} \quad u \leftrightarrow o \leftrightarrow \mathfrak{o} \leftrightarrow a.$$

The problem is designed under the assumption that the solver can differentiate the front unrounded vowels (left) from the back vowels (right). This would be expected knowledge of a student sitting the UKLO R2 paper.

In both cases, an arbitrary new orthography would render the problem nearly impossible to solve, since it would not visibly preserve the important linguistic properties of the sounds that the graphemes represent.

---

[5]System for writing a language, including the choice of letters or other glyphs and spelling conventions.
[6]Letter or group of letters that typically represent a single sound or other feature.

To prevent issues of type (i), we manually found all cognates in the Problemese data, and ensured that they were not respelled if it would affect the difficulty of the problem. In cases of cognates that had no impact on the problem, such as most problems about European languages, these cognates were still respelled. We also ensured that all names of people, deities, and sacred places were not re-spelled.

To prevent issues of type (ii), we created problem-specific rulesets describing which permutations of the graphemes would not affect the solvability or difficulty of the problem. We call such permutations *valid*. Our permutation rulesets are highly problem-specific, only taking into account linguistic properties of sounds that were relevant to solving that problem. Aside from problems where there were no such properties, no two problems had the same set of relevant linguistic properties, and so each problem's permutation ruleset is unique. In Section B.5, we detail the construction of these permutation rulesets for two problems.

Sampling permutations according to these rulesets allows for the dynamic generation of new obfuscated variations of the problems in the benchmark.

## B.2 METHODOLOGY

To determine the permutation ruleset for a given problem, the following procedure was applied.

First, the set of graphemes present in the Problemese data was determined. This involved using the authors' expert judgement on which strings should be analysed as single graphemes, and which could be split into multiple units. A common example was deciding when a combination of a letter with a diacritic should be treated as a single grapheme (as in Spanish $ñ$) or as multiple graphemes (as in Spanish $á = a + ´$ ). This sometimes involved consulting literature on the language, but usually it was sufficient to consult the commentary on the problem provided by UKLO, as well as the authors' own solutions to the problems. Note that a grapheme could be a substring of another grapheme – consider English $h$ as a substring of $sh$. In this situation, the permutation algorithm would scan the annotated text left-to-right, and exchange the longest matching grapheme with its replacement under the obfuscation in a greedy manner.

We then determined what phonetic and phonological data needed to be preserved in the orthography for the problem to remain solvable and the difficulty to remain unchanged. In many cases, more phonological data was preserved than strictly necessary, such as always keeping vowels and consonants separate. This was both to preserve alternative solving paths than the authors', which may have involved a different set of phonological observations, and to allow for consistent syllabification across permutations. This data was then encoded in the permutation ruleset for the problem, for example by separating out certain sounds to only be permuted amongst themselves, such as with $\{s, z, š, ʒ\}$ in Tadaksahak, or by fixing especially difficult graphemes.

Whenever valid permutations were sampled to generated obfuscated versions of problems, the permutation was chosen to have no fixed points outside of the fixed set. This was achieved by selecting a random cycle whenever a random permutation of a set of graphemes had to be chosen.

## B.3 STRUCTURE OF A PERMUTATION RULESET

A permutation ruleset for a particular problem partitions the graphemes in the Problemese data into four types of structured collection: *sets*, *tables*, *free-tables* and a *fixed set*. A permutation is a choice of structure-preserving bijection of each of these collections, which combine to give a bijection of all of the graphemes in the data.

**Sets**    A set is simply a collection of graphemes, with no extra structure that needs to be preserved. A permutation can therefore be any bijection of the graphemes in a set.

**Tables**    The graphemes in a table are partitioned into tuples, each of the same length, thought of as columns of a table. The permutation then freely permutes these 'columns', while preserving the 'rows' (index in the tuple) where the graphemes appear. For example, a table

$$\{(p, b), (t, d), (k, g)\} = \begin{array}{|c|c|c|} \hline p & t & k \\ \hline b & d & g \\ \hline \end{array}$$

will have $3! = 6$ ways to rearrange the 3 columns, hence 6 possible permutations:

$$
\begin{array}{|c|c|c|}\hline p & t & k \\\hline b & d & g \\\hline\end{array}
\xrightarrow{\text{obfuscations}}
\begin{array}{|c|c|c|}\hline p & t & k \\\hline b & d & g \\\hline\end{array},\;
\begin{array}{|c|c|c|}\hline t & p & k \\\hline d & b & g \\\hline\end{array},\;
\begin{array}{|c|c|c|}\hline k & t & p \\\hline g & d & b \\\hline\end{array},\;
\begin{array}{|c|c|c|}\hline p & k & t \\\hline b & g & d \\\hline\end{array},\;
\begin{array}{|c|c|c|}\hline t & k & p \\\hline d & g & b \\\hline\end{array},
$$

$$
\begin{array}{|c|c|c|}\hline k & p & t \\\hline g & b & d \\\hline\end{array}
$$

Linguistically, this example represents a situation where plosives appear in voiceless/voiced pairs, and this data must be preserved, but the place of articulation of the plosives is not relevant, so may be permuted. Compare this to a situation where only the voicedness was relevant, but not the voiceless/voiced pairs, which could be obfuscated by a pair of sets $\{p, t, k\}$, $\{b, d, g\}$, allowing for $3! \times 3! = 36$ possible obfuscations.

**Free-Tables**    These are a generalisation of tables where some of the 'rows' consist of collections of more than one element, together forming a set. A choice of permutation is first a choice of bijection of the columns, then a choice of bijection from each original 'cell' to the obfuscated 'cell'. For example, a free-table

$$
\{(m, \{p, b, f\}), (n, \{t, d, s\})\} = 
\begin{array}{|c|c|}\hline m & n \\\hline p,\,b,\,f & t,\,d,\,s \\\hline\end{array}
$$

has $2! \times (3!)^2 = 72$ possible permutations. An example is shown below, where the non-identity bijection of the columns was chosen.

$$
\begin{array}{|c|c|}\hline m & n \\\hline p,\,b,\,f & t,\,d,\,s \\\hline\end{array}
\xrightarrow{\text{obfuscation}}
\begin{array}{|c|c|}\hline n & m \\\hline t,\,s,\,d & b,\,p,\,f \\\hline\end{array}
\qquad
\overset{\cong}{\longleftrightarrow}
\qquad
\begin{array}{ll}
m \mapsto n & n \mapsto m \\
p \mapsto t & t \mapsto b \\
b \mapsto s & d \mapsto p \\
f \mapsto d & s \mapsto f
\end{array}
$$

Linguistically, this represents a situation where homorganic nasals are relevant, but otherwise place and manner of articulation are not relevant.

**Fixed Set**    Any grapheme not in any set, table or free-table was fixed by all of our permutations. Additionally, other strings that should not be changed during obfuscation could be specified as a single grapheme within the fixed set. Such strings primarily fell into two categories: transparent loanwords or cognates, and names. Where words in the Problemese data had noticeable resemblances to English or another well-known language, such as Nepali *pēnsil* 'pencil' or Yawalapiti *amiku* 'friend' (cf. Portuguese *amigo*), and where these resemblances may have decreased the difficulty of a the problem by providing a starting point for reasoning, they were always preserved. The only times that transparent cognates were not preserved was when the problem did not involve the meanings of words, so recognition of the cognate would provide no advantage. Names of people (both real and fictional), deities, and sacred places were also always preserved. The names of other places and languages were preserved where possible, but not if it would consequently indicate what the Problemese language was.

## B.4    COMPARISON TO OBFUSCATION IN LINGUISTICS OLYMPIADS

This method of obfuscation is more severe than what is typically used by Linguistics Olympiads. Most Linguistics Olympiads do not perform any intentional obfuscation, but where it is done common notations are exchanged for other common notations as far as possible. For example, in UKLO 2021 A4 (Sauk by Ryan Chi), long vowels were changed from being written with a circumflex (e.g. *â*) to being doubled (e.g. *aa*). This is typically sufficient to prevent search engines from returning relevant results, but has little effect on any advantage gained by having prior familiarity with the language.

## B.5    EXAMPLES OF PERMUTATION RULESETS

We present the thought process behind the creation of the permutation rulesets for two problems, Somali (Problem 91, by Harold Somers) and Stodsde (Problem 218, by Simi Hellsten). These are representative of the process behind all of the more complicated permutation rulesets. We first

give a simplified form of the grammar of the language that a solver would be expected to deduce, then explain how this solution could be used to determine a permutation ruleset. Note that expert judgement, coming from familiarity with both these specific problems and Linguistics Olympiads in general, is required to construct these rulesets.

**Problem 91, Somali** (by Harold Somers). This problem involves determining the formation of 1sg and 3sg forms of Somali[7] verbs. The solution is as follows:

   (i) The 1sg form adds *-ay* to the root, while the 3sg form adds *-tay*.

   (ii) After the consonants *q, c, x, '* (the so-called guttural consonants), we have the change $t \to d$.

   (iii) Everywhere, we have the following changes: $yt \to d$; $lt \to sh$; $dt \to d$; $dht \to dh$.

There are no names or loanwords in the Somali data that need to be preserved by the obfuscation, so the permutation ruleset will only depend on the phonology of the problem. All permutations should fix *t, d, dh, sh*, since they have phonological relationships which are relevant to the solution of the problem, and cannot be found in a different family of consonants. The guttural consonants can be permuted amongst themselves but not with the other consonants, so form a set $\{q, c, x, '\}$. *l* and *y* are treated specially within the solution, but any liquids could fill those roles without substantially changing the difficulty of the problem, so we define a set $\{l, r, w, y\}$. All other consonants featured in the Somali data form a set $\{b, f, g, j, h, k, n, s\}$. The vowels form another set $\{a, e, i, o, u\}$.

**Problem 218, Stodsde** (by Simi Hellsten). This problem involves understanding the formation of causative verbs in Stodsde.[8] The solution involves understanding the structure of a Stodsde syllable, and the changes that occur in each part of the syllable. No changes occur in the vowel, or consonants after the vowel. The part of a syllable before the vowel has the following structure, divided into Slots 1–5. Each slot can only have consonants of a certain type, and undergoes specific changes to form the causative.

| Slot 1 | Slot 2 | Slot 3 | Slot 4 | Slot 5 |
|---|---|---|---|---|
| Nasal | Non-sib. fricatives | Liquid (*l* or *r*) | plosive/sibilant | Any non-plosive + non-sib. |
| $m \to v$ $n \to \emptyset$ | | $\to \mathrm{z}$ if Slot 4 is sibilant $\to z$ otherwise | fricative $\to$ affricate, unless Slot 5 is a nasal | |

Finally, everywhere in the verb, $v, \mathrm{z}, z \to f, \mathit{ł}, s$ if the following sound is voiceless. Several alternative and equivalent analyses are also possible.

Again, the dataset contains no loanwords or names, so the permutation ruleset is fully determined by the relevant phonology. Since voicedness is relevant throughout, but not all voiced/voiceless pairs are given in the data, any permutation must preserve voicedness. We also fix $\mathrm{z}$, hence fix *ł*. Since we cannot permute $\emptyset$, we must fix all of Slot 1. Thus *m, n*, and *v* are fixed, hence also *f*. For Slot 2, the remaining non-sibilant fricatives are the voiceless $\chi$, which cannot be permuted with anything, and voiced $\text{\textpm}, \gamma$, which can be permuted. In Slot 3, we can freely permute *l, r*. Slot 4 needs to distinguish between sibilants and sibilant affricates (which are paired), and simple plosives. Considering only those that appear in the Stodsde data, this means we must fix $ts^h$; we can permute *p, k, q*, and *b, d, g*, while fixing the aspiration $^h$; and we can permute the pairs $(z, dz)$, $(\mathfrak{z}, d\mathfrak{z})$, and the pairs $(s, ts)$, $(s^h, ts^h)$. Finally, we can permute the vowels, and the remaining nasals $\eta, \textipa{\textltailn}$; *j* has nothing it could permute with, so must also be fixed.

The final permutation ruleset thus has fixed set $\{m, n, \mathit{ł}, \mathrm{z}, \chi, v, f, ts^h, {}^h, j\}$; sets $\{\text{\textpm}, \gamma\}$, $\{l, r\}$, $\{p, k, q\}$, $\{b, d, g\}$, $\{\eta, \textipa{\textltailn}\}$, $\{u, \textschwa, \textturna, o, a, æ, i\}$; and tables $\{(z, dz), (\mathfrak{z}, d\mathfrak{z})\}$, $\{(s, ts), (s^h, ts^h)\}$.

---

[7]Somali is a Cushitic language spoken by approximately 24 million people, primarily ethnic Somalis in East Africa and in diaspora.

[8]Stodsde is a Gyalrongic language, spoken by approximately 4,000 people in Sichuan, China.

| Problem | Original | Annotated |
|---|---|---|
| 67 | This problem is about the way in which Navajo speakers build sentences out of a verb V. | This problem is about the way in which $$$Language X$$$ speakers build sentences out of a verb V. |
| 16 | Ulwa is a language spoken in Nicaragua. It contains quite a few loanwords from English, which is spoken in the Bluefields area of the country. | $$$Language X$$$ &&& &&& contains quite a few loanwords from English &&& &&&. |
| 61 | dinaldalusanda they were cleaning it | @@@dinaldalusanda@@@ they were cleaning it |
| 61 | t(in)ak+takaw+da ida | tinaktakawda ida |

Table 5: **Annotation examples.** Each row is an extract from a problem in LINGOLY-TOO. On the left, the extract appears in the original UKLO problem sheet or solution file. On the right is the extract after annotation. *Problem 67* (Navajo by Babette Verhoeven) is an example of the language name annotation. *Problem 16* (Ulwa by Richard Sproat) is example of the cultural context annotation. *Problem 61* (Ilokano by Bozhidar Bozhanov) illustrates the annotation of Problemese in preparation for further obfuscation, and is an example of removing grading guidelines from the dataset to prepare it for checking exact matching.

## C  ANNOTATIONS

### C.1  ANNOTATION PROCESS

Annotation was the first step of the obfuscation process in LINGOLY-TOO. The primary task was removing metadata included in the problem text such as language names, language families, and geographic information that was not relevant for solving the problem via reasoning, but could have allowed models to ascertain facts about the language, even when obfuscated. Problemese data that would be altered in later stages of obfuscation were also highlighted during annotation.

All problem annotations were performed by a single annotator to minimise the inconsistency between different problems. Two other members of the team had expertise in Linguistics Olympiads. They validated each of the annotated problems, identifying and documenting any errors or inconsistencies that needed correcting. In addition, all annotations were parsed through automatic checks implemented as part of the data generation pipeline. Despite our best efforts to ensure all problems are all correct, mistakes may remain as the result of undetected human error.

### C.2  ANNOTATION CATEGORIES

Four categories of annotations were performed: language and place names, cultural context, Problemese data, and grading guidelines. Each involved bracketing strings in the problem files with triplets of a symbol not present or used elsewhere in the problems.

**Language and Place Names**    The English name of the Problemese language was replaced with 'Language X' wherever it appeared in the problem, and was annotated using $$$ tag (see row one in Table 5). If multiple non-English languages were mentioned in the problem, the second was replaced with 'Language Y' and the third with 'Language Z'. A maximum of three languages appeared in any given problem. Where related terms such as geographic regions or people groups associated with those languages were relevant to the problem, they were replaced with e.g. 'Region X', and also annotated using the $$$ tag. Where two historical varieties of the same language were featured in the same problem, their names were replaced with 'Language X' and 'Language Y', and a sentence clarifying their relationship was added.

**Cultural Context**    Any other metadata or cultural references that could indicate the origin of a language but were not relevant to solving the problem were replaced with a space, and annotated with

&&& tag (see row two in Table 5). If it was not clear whether some part of the metadata might be useful in solving the problem, it was annotated with the $$$ tag, and edited in accordance with the principles in the previous paragraph.

**Problemese Data** All Problemese data were annotated using an @@@ tag (see row three in Table 5). Where problems involved multiple languages, these were not distinguished in annotation. Instead, the permutations of graphemes used to alter the Problemese data were designed to preserve the distinctions between the languages. Many of the problems included pronunciation guides, since students sitting UKLO exams are not expected to have any prior linguistic knowledge. These pronunciation guides were not annotated, since the permutations of the graphemes used to alter the Problemese always preserved any relevant phonetic and phonological distinctions.

**Grading Guidelines** Most UKLO answer sheets include marking guidelines for awarding partial credit, which are often accompanied by symbols such as '+', '|' and brackets written inside the correct answers. These were simply removed from the answer files (see row four in Table 5).

## D PROBLEM DIFFICULTY BREAKDOWN

| Difficulty Level | Unobfuscated | Obfuscated |
|---|---|---|
| Breakthrough | 60 (6.0%) | 320 (5.3%) |
| Foundation | 84 (8.4%) | 504 (8.4%) |
| Intermediate | 136 (13.5%) | 816 (13.6%) |
| Advanced | 374 (37.2%) | 2244 (37.5%) |
| Round2 | 351 (34.9%) | 2106 (35.2%) |

Table 6: **Counts (percentages) by difficulty level.** This is for the $1,005$ unobfuscated (sub-question, answer) pairs and the $5,990$ obfuscated (sub-question, answer) pairs. Note that the obfuscated percentages are slightly different compared to the unobfuscated as some problems allow for less than 6 permutations.

## E PROMPT TEMPLATE

Below is the prompt used across models. In the *no context* setting, {context} is removed.

```
Below is a problem sheet from a linguistics exam. You will first see the
    entire sheet, then be asked to respond to specific questions from
    the sheet. Your answers to the questions should rely only on
    reasoning about the information provided in the sheet.
{question_body}

Now respond to the following questions:
{preamble}
{context}
{all_subquestions}

{instructions}
{formatted_output}
```

The {formatted_output} tag is populated with an empty dictionary containing the keys of the expected answer for each part of the question. The {instructions} tag is populated based on the setting. For default prompt:

```
Only respond with json output. Do not include anything other than the
    json in your response. Format your response as a json file with the
    keys as provided below:
```

in closed models when applicable, we preceded the prompt with the following generic system message:

```
You are a helpful assistant.
```

## F  EVALUATION MODELS

In our experiments, we fixed prompts and system messages across models whenever possible. For closed models, we followed the recommendations of model providers and set the temperature to $0$ whenever possible. For open models, we loaded them in full precision when possible or in 8-bit for larger models. All evaluation experiments were conducted using nodes with either 4 A100 GPUs or 4 H100 GPUs for open source models except for Deepseek R1 [9]. Model details are listed listed in Table 7.

| Model | Version | Type | Quantised |
|---|---|---|---|
| Claude 3.5 Sonnet | claude-3-5-sonnet-20241022 | Proprietary | |
| Claude 3.7 Sonnet (thinking) | claude-3-7-sonnet-20250219 | Proprietary | |
| Claude 3.7 Sonnet (no thinking) | claude-3-7-sonnet-20250219-nothinking | Proprietary | |
| Claude Opus 4.1 | claude-opus-4-1-20250805 | Proprietary | |
| DeepSeek-V3.1-Terminus | deepseek/deepseek-reasoner | Open-source | |
| Gemini 1.5 Pro | gemini-1.5-pro | Proprietary | |
| Gemini 2.5 Pro | gemini-2.5-pro-exp-03-25 | Proprietary | |
| GPT-4o | gpt-4o-2024-05-13 | Proprietary | |
| GPT-4.5 | gpt-4.5-preview-2025-02-27 | Proprietary | |
| GPT-5 | gpt-5-2025-08-07 | Proprietary | |
| Llama 3.3 70B | meta-llama/Llama-3.3-70B-Instruct | Open-source | 8-bit |
| o1-preview | o1-preview | Proprietary | |
| o3-mini | o3-mini-2025-01-31 | Proprietary | |
| Phi4 | microsoft/phi-4 | Open-source | 8-bit |

Table 7: **Details of models evaluated.**

## G  SUMMARY OF RESPONSE ERRORS

Table 8 contains the summary of responses for all prompts by model. A higher number of responses by o3-mini (low) were empty due to generated reasoning tokens exceeding the maximum token budget but excluding empty responses by o3-mini (low) did not affect its ranking.

---

[9]For Deepseek R1, we used https://www.together.ai/models/deepseek-r1.

| Model | Total | Empty Response | Bad Parsing |
|---|---|---|---|
| Claude 3.5 Sonnet | 6,995 | 8 | 0 |
| Claude 3.7 Sonnet (no thinking) | 6,995 | 1 | 0 |
| Claude 3.7 Sonnet (thinking) | 6,995 | 22 | 0 |
| Claude Opus 4.1 | 6,995 | 7 | 0 |
| DeepSeek-V3.1-Terminus | 6,995 | 115 | 0 |
| GPT-4.5 | 6,995 | 0 | 6 |
| GPT-4o | 6,995 | 176 | 0 |
| GPT-5 | 6,995 | 225 | 0 |
| Gemini 1.5 Pro | 6,995 | 16 | 0 |
| Gemini 2.5 Pro | 6,995 | 19 | 0 |
| Llama 3.3 70B-Instruct | 6,995 | 316 | 0 |
| Phi4 | 6,995 | 287 | 0 |
| o1-preview | 6,995 | 148 | 0 |
| o3-mini (high) | 6,995 | 38 | 11 |
| o3-mini (low) | 6,995 | 1,208 | 10 |

Table 8: **Summary of model responses for benchmark prompts.**

## H  FULL RESULTS

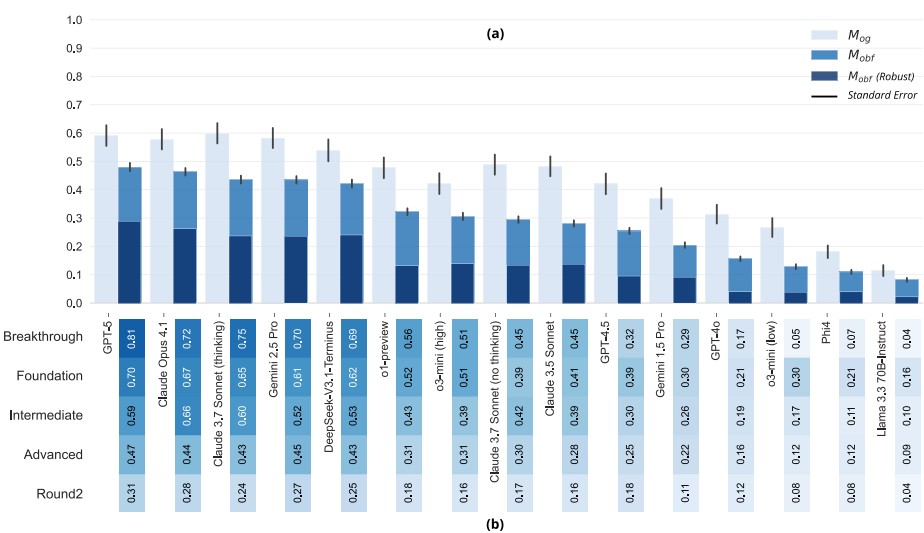

Figure 6: **Main results on LINGOLY-TOO.** (a) Scores by model. $M_{og}$ is based on the original problems and $M_{obf}$ is based on the obfuscated problems. $M_{obf}$ (Robust) is calculated after taking the worst score across all permutations of the question. (b) Breakdown of $M_{obf}$ scores by difficulty level.

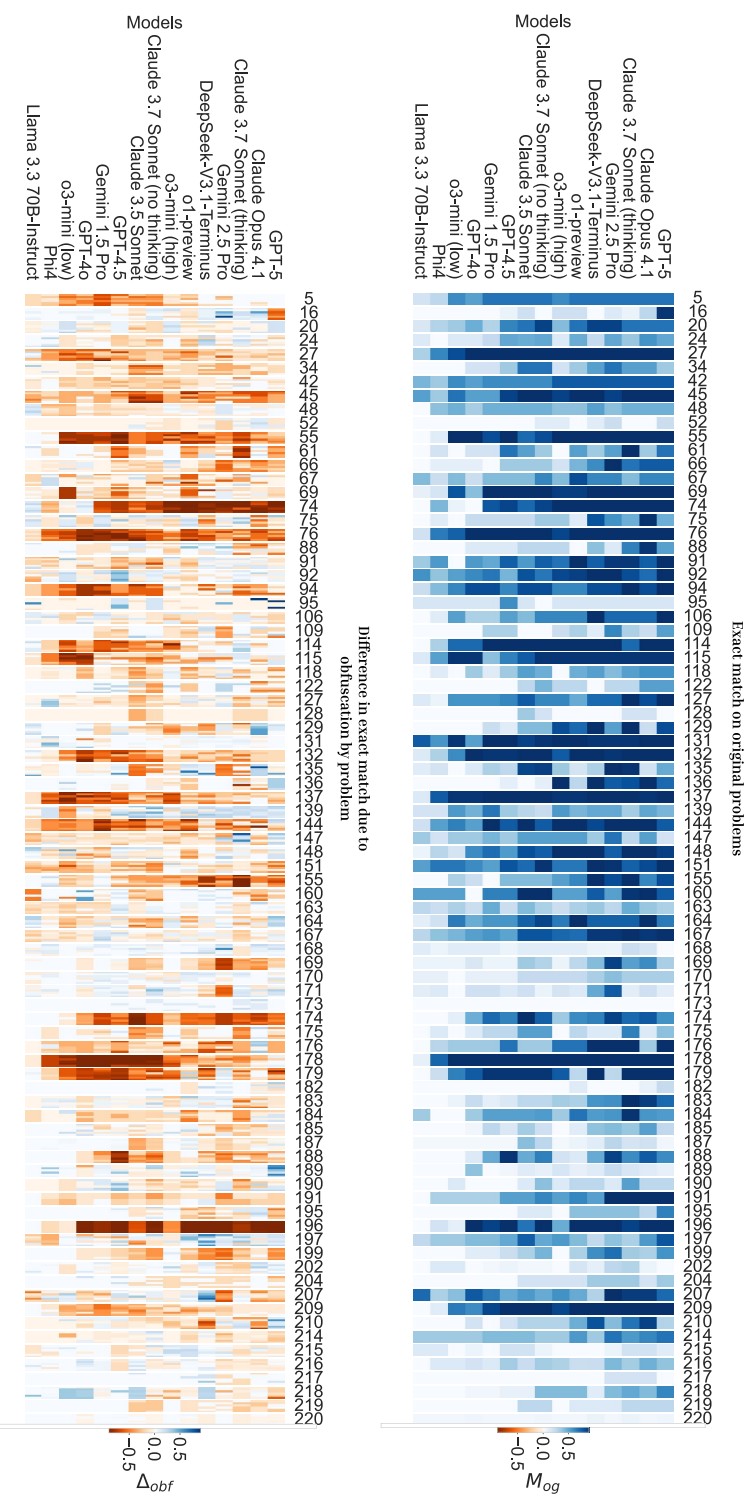

Figure 7: **Heatmap of results on each problem for each model.** Right: the scores on the unobfuscated problems. Left: The differences between the score on the unobfuscated problems and the average of the obfuscated versions. Red indicates a lower performance after obfuscation.

# I SCORE DISTRIBUTION DETAILS

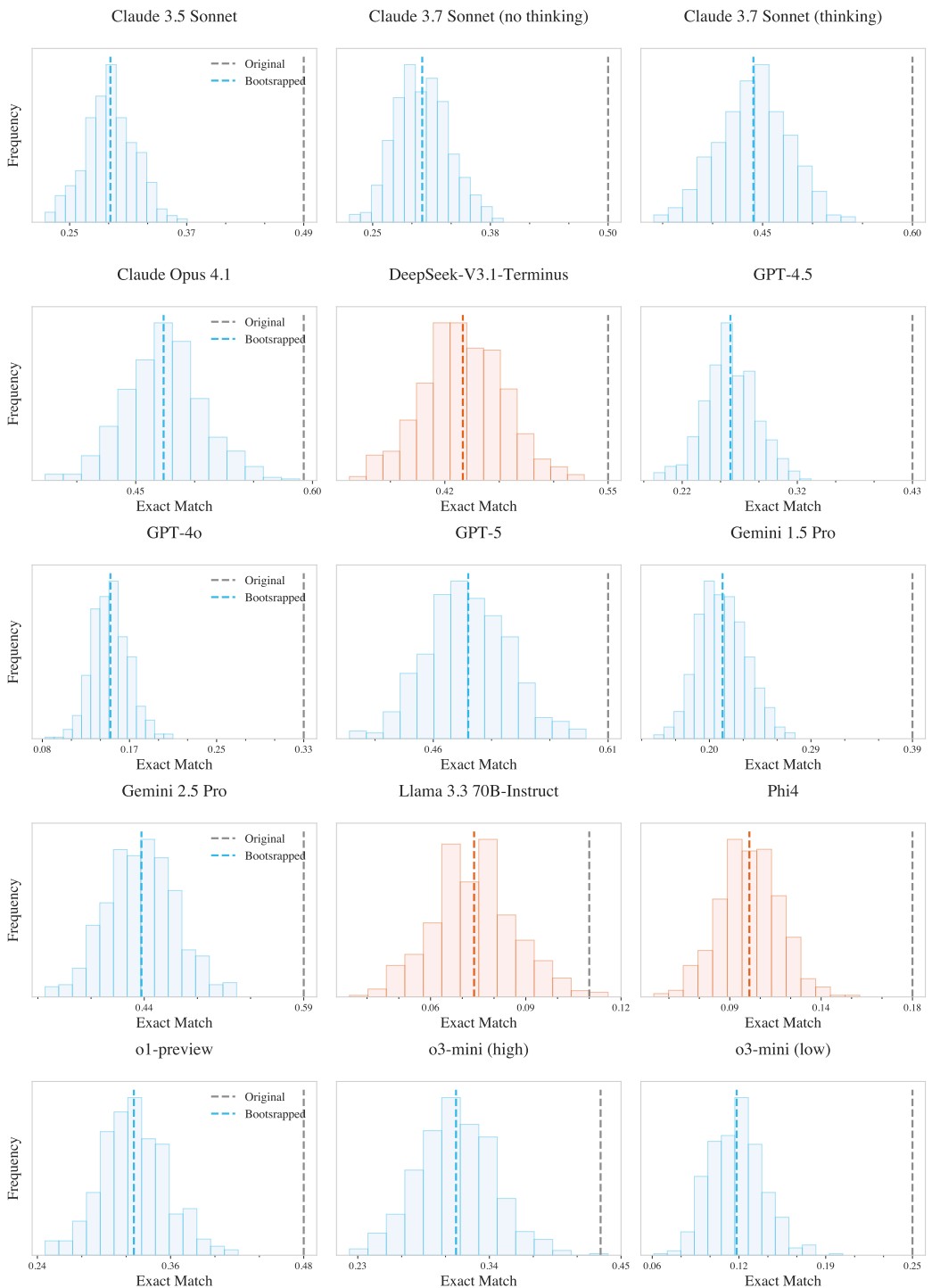

Figure 8: **Distribution of bootstrapped scores across 500 bootstrapped samples over 82 problems.** Open source models are shown in orange while proprietary models are in blue. The performance on the original, unobfuscated problems consistently appear at the right tail.

## J  UKLO 2025 Questions

| Model name | Unobfuscated | Obfuscated |
|---|---|---|
| Claude 3.7 Sonnet (thinking) | 0.25 | 0.20 |
| Claude 3.5 Sonnet | 0.25 | 0.15 |
| GPT 4o | 0.16 | 0.09 |
| o3-mini (high) | 0.10 | 0.08 |

Table 9: **Performance by model on unobfuscated vs obfuscated questions from the UKLO 2025 paper.** Note that this paper was released after these models have been trained therefore the performance gap cannot be attributed to training set contamination.

We were able to access, annotate, and evaluate a subset of models on 5 problems from the UKLO 2025 papers that have not yet been published online by the time of the experiment and are of a higher difficulty level. Results in Table 9 are comparable to results on the benchmark. The difference in scores illustrates that performance gap is not only due to memorising answers since these problems have not been included in models training. The gap also highlights the effectiveness of our permutation to control for prior knowledge even on unseen problems, which is the main aim of the benchmark.

## K  Tokenisation Experiment

We conduct an experiment to explore if the impact on tokenisation due to permuting the graphemes of a language explains the gap in model performance.

### K.1  Set-up

We choose the Aya 23 35B model by Cohere (Aryabumi et al., 2024), which was designed for multilingual applications (23 languages). The tokeniser of Aya 23 is based on the popular byte-pair encoding algorithm (BPE) (Gage, 1994). Since we report off-the-shelf model performance, we investigate changes in the performance if we alter the tokenisation of the Problemese portion of the prompt.

When prompting this model, the Problemese sections are treated separately. In the first experiment, we break down the Problemese into unicode characters (with the NFD standardisation) and obtain separate tokenisation for each character that we then concatenate together. In the second experiment, we add dashes in-between the characters of the Problemese to force some separation in the interpretation of multiple characters. The rest of the question is tokenised in the standard way, and the problem context and the Problemese are then concatenated together (in the order that they appear in the question) before being fed into the model.

For this experiment, we used 10 obfuscations of all problems except Problem 34 (which had long prompts taking up too much memory).

We limited the model outputs to four times the length of the correct response length. After cleaning, the number of incorrectly processed responses is 32 questions for the standard tokenisation, 289 questions for the input with dashes, and 292 questions for the single character tokenisation (out of a total of $10,765$ questions). This may partially account for the additional drop in the performance of the alternative tokenisations.

### K.2  Additional results

In Table 10 we see how the average model performance changes under alternative tokenisation for the unobfuscated problems. For the unobfuscated case (Table 10), the performance decreased for a greater number of problems than for the obfuscated case (Table 11). This aligns with the greater drop in performance that we observe in Table 3.

Finally, we show how the exact match scores vary under alternative tokenisations in Figure 9 by plotting (standard tokenisation score, alternative tokenisation score) for each problem. When the

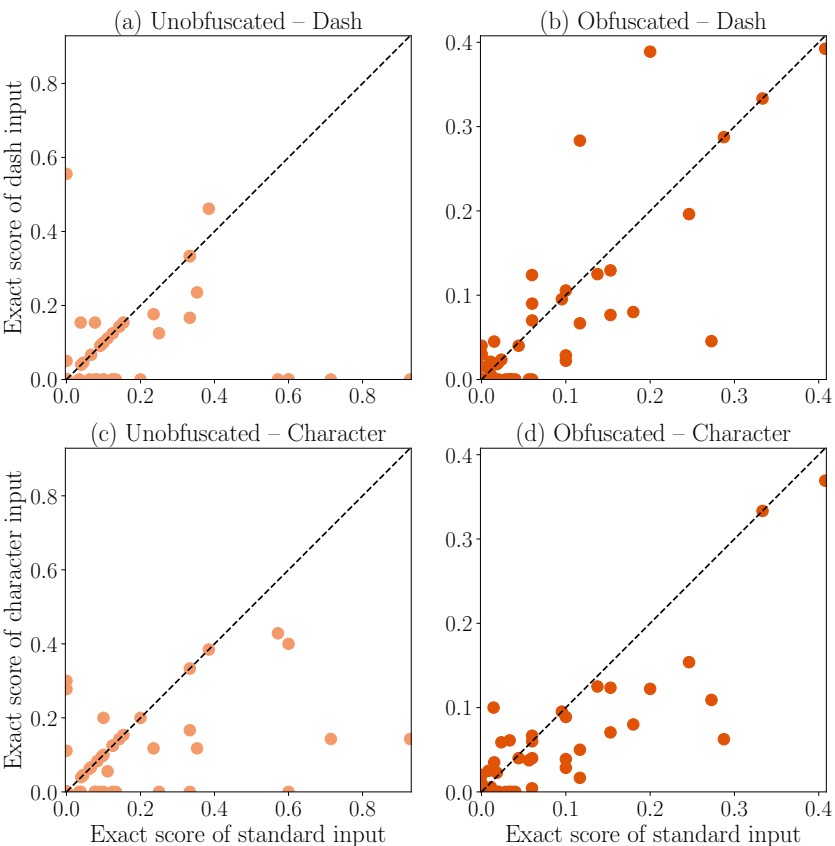

Figure 9: **Problem-level comparison of exact match scores between standard and alternative tokenisation.** The dashed line represents the threshold where the score remains unchanged after altering the tokenisation. Point above the dashed line indicate better performance with alternative tokenisation, while points below the line indicate worse performance. (a) The problems are *unobfuscated* and we compare scores of standard tokenisation against *dash* tokenisation. (b) The problems are *obfuscated* and we compare scores of standard tokenisation against *dash* tokenisation. (c) The problems are *unobfuscated* and we compare scores of standard tokenisation against *character-level* tokenisation. (d) The problems are *obfuscated* and we compare scores of standard tokenisation against *character-level* tokenisation.

| Prompt type | Decreased | Unchanged | Increased |
|---|---|---|---|
| Dash | 57 | 19 | 5 |
| Character | 58 | 19 | 4 |

Table 10: **Performance change under alternative tokenisation for the unobfuscated problems.** Score for each problem was averaged over all questions within the problem for the two alternative tokenisation methods (Dash and Character), then compared to the score obtained using standard tokenisation and categorised into Increased, Decreased, or Unchanged.

| Prompt type | Decreased | Unchanged | Increased |
|---|---|---|---|
| Dash | 43 | 25 | 13 |
| Character | 37 | 30 | 14 |

Table 11: **Performance change under alternative tokenisation for obfuscated problems.** For each alternative tokenisation, the score for each problem was averaged over all obfuscations then compared to the score obtained using standard tokenisation and categorised into Increased, Decreased, or Unchanged.

problems are not obfuscated (Figure 9(a) and (c)), the alternative tokenisation more frequently reduced the score to zero, indicating that the tokenisation of the original order may be capturing useful information. For the obfuscated cases (Figure 9(b) and (d)), there appeared to be more spread in the scores differences, and no instances of the catastrophic effect of changing from a high exact match score under the standard tokenisation to a score of zero.

### K.3 SUMMARY

We explored whether the drop in performance could be explained by tokenisation of rare character sequences rather than reduced access to prior knowledge. Using a multilingual model (Aya 23 35B) with BPE tokenisation, we compared standard tokenisation to (i) inserting a dash between every character in Problemese and (ii) forcing single-character tokens. The exact-match score did not improve under either alternative: on unobfuscated problems $0.087$ against $0.051$ or $0.053$, and on obfuscated questions $0.050$ compared with $0.045$ or $0.035$. This argues against tokenisation alone as the driver of the performance gap: for each type of tokenisation, there was a marked drop in the score after obfuscation.

Problem-level analyses show many more decreases than increases, especially on unobfuscated problems, consistent with the view that original orthographies benefit from familiar segmentations while obfuscated orthographies remove these knowledge shortcuts. Taken together with "no-context" results, these findings support our interpretation that obfuscation of these linguistic problems mitigates predicting the answers from prior knowledge.

While it is true that the tokenisation effect may not be the same across problem domains, tokenisation has also been shown to make a difference in other domains like arithmetic problems (Singh & Strouse, 2024). We expect that the direction of our finding to hold more generally: we introduce logically equivalent permutations and show that LLMs are not equivariant under these permutations in a way that is unexplained by only tokenisation effects.

### L HUMAN EVALUATION

Obfuscation does not change the reasoning steps required to solve the problem, and therefore it serves as a method to robustly test reasoning capabilities whilst mitigating memorisation bias. However, obfuscation may have an effect on how humans solve the problems through means other than changing the reasoning process. To measure the effect of obfuscation on human performance, we carried out a randomised controlled trial (RCT) with 172 human participants across 6 problems (all relatively easy). After controlling for random differences in problem frequency, obfuscation was found to be associated with $5.80$ percentage points lower performance ($p$-value of $0.059$). We speculate that

this may be because the resulting character combinations are more unusual in naturally occurring languages, thus the problems are perceived as being harder.

## L.1 METHODOLOGY

**Experiment Design** We used an RCT with 172 participants. Each participant was assigned to one of two groups: one group solving unobfuscated problems (86 people) and another solving obfuscated problems (86 people). Participants were then assigned one of six problems and given up to 45 minutes to complete the problem. They were required to spend a minimum of 30 minutes on the problem before submitting their answers. Responses were scored using exact match with the answers. All data collection was carried out through a Qualtrics survey.

**Problem Inclusion Criteria and Design** We used two criteria to select problems. First, we only considered *Breakthrough* and *Foundation*-level problems, designed to be taken by 10–14 year olds. Our intention was to ensure participants had a realistic chance of solving the problem, and thus the responses would provide better signal of the effect of obfuscation. Second, we only considered problems where the language was not available through Google Translate, since this would have enabled participants to easily solve the problems. Two exceptions to this rule were Karelian, which is not available on Google Translate but was removed due to a high similarity with Finnish, and Ligurian, which is available on Google Translate without stress being indicated, which formed the basis of the problem. The selected six problems are below.

| Language | Difficulty level | Problem | Original author(s) |
|----------|------------------|---------|--------------------|
| Warlpiri | Breakthrough | 207 | Mary Laughren |
| Umbrian | Breakthrough | 191 | Michael Salter |
| Kabyle | Breakthrough | 160 | Kazune Sato, Simi Hellsten |
| Tariana | Foundation | 210 | Babette Verhoeven, Simi Hellsten |
| Ligurian | Foundation | 147 | Kevin Liang |
| Amele | Foundation | 88 | Babette Verhoeven |

Table 12: **Problems selected for the randomised controlled trial.** All problems are *Breakthrough* and *Foundation*-level, designed to be taken by 10–14 year olds. None of these languages are available on Google translate in a manner that would aid the solvability of the problem. Problem is the listing of the problem in the UKLO past problems database (United Kingdom Linguistics Olympiad, 2023).

Since all six problems and their solutions are easily accessible on the UKLO website, we rewrote elements of each problem. First, we rephrased the problem instructions so that a quick internet search did not return the solutions. Second, we changed some of the lexemes in the Problemese such that the linguistic rules required to answer the questions were identical, but the specific words to translate were different. For example, in Kabyle we substituted the verb *azzel* 'to run' with *aker* 'to steal', which follows the same conjugation pattern. All problems were minimally rewritten except from Umbrian, where the known vocabulary originates exclusively from inscriptions on the Iguvine Tablets, and thus is extremely small. Additionally, as in the LLM experiments, all background and cultural information was removed, and the language name was replaced by 'Language X'.

For each problem, we randomly sampled two permutations. Participants who were assigned to the obfuscated group were randomly assigned one of the two obfuscated versions.

**Participant Inclusion Criteria** Participants were recruited online using Prolific, a crowdworker platform. First, all participants had to be English speaking and monolingual, reducing the risk that they may have had prior exposure to the Problemese language. Second, all participants were required to have a minimum education level of an undergraduate degree. This served as a proxy for ability and was chosen to ensure participants could reasonably complete the problems.

Participants were compensated in line with the UK aged-21 and over minimum wage of £11.44 per hour (experiments conducted January 2025). Additionally, participants received performance bonuses of £2.00 if they scored above 50% but less than 100%, and £6.00 if they scored 100%. The incentive structure was explained to participants before they started the problem.

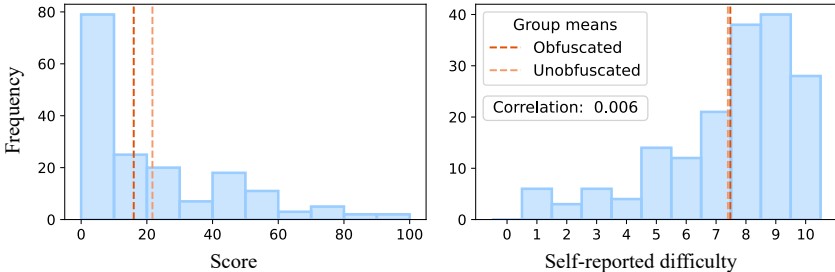

Figure 10: **Distributions of participant scores and self-reported difficulty.** *Left:* Distribution of scores across all six problems and problem types. Scores are highly skewed towards 0. The original group had a higher mean score compared to the permuted group. *Right:* Participant self-reported difficulty. Participants were asked the question *'On a scale from 0 (very easy) to 10 (impossible), how challenging did you find this problem?'*. Responses are highly skewed towards 10 (impossible). The permuted and original means are near identical. Despite the symmetric distributions, correlation between score and self-reported difficulty is 0.006.

All ICML Publication Ethics protocols were followed and the experiment had prior approval by the research ethics committee at our institution. All participants were required to provide informed consent by signing the form shown in Section L.5.

**Controlling for Internet and LLM Use**  To control for the possibility that participants search for the solutions, each question was presented in image form, making it harder to directly copy and paste questions into an internet browser. Additionally, we added a post-response *trap question* asking *'What do you think the language in the problem (Language X) was?'*. Given the obscurity of the chosen languages, we assumed a correct answer to this question implied that participants had either found the solutions or were using a language model, both of which were explicitly prohibited in the instructions.

To mark responses to the trap question, we first automatically identified all entries which identified the correct language. Next, two authors with significant language model experience independently identified all responses which qualitatively appeared AI generated. Evidence suggests that people who frequently use state-of-the-art LLMs are good detectors of AI generated text, often outperforming commercial detection software (Russell et al., 2025). These included responses containing common language model phrases, excessively long responses and those demonstrating an unusually impressive knowledge of academic linguistics, such as claiming the Problemese was likely an *agglutinative language*. Each author independently marked responses, then discussed cases where they disagreed. Since suspected cheating in the trap question does not necessarily imply that the participant cheated in the main problem, these participants were left in the data for the main analysis. Results excluding these participants are reported as a robustness check.

### L.2 RESULTS AND ANALYSIS

**Overall Performance**  Participants found these problems hard, with mean performance across all problems at 18.85% (mean random guessing is 9.19%). Responses to the question *'On a scale from 0 (very easy) to 10 (impossible), how challenging did you find this problem?'* reveal that many participants found the problems near impossible, with 39.5% reporting a difficulty of 9 or 10 out of 10. There is no significant difference in the distributions of self-reported difficulty across the participants taking obfuscated and unobfuscated problems (Figure 10).

Table 13 shows the mean score by problem and obfuscation type for both human participants and the LLMs. The mean score for human participants solving the unobfuscated and obfuscated problems were 21.70% and 16.00%, respectively. To assess the statistical significance of this change, we used a one-sided Mann–Whitney U test. This was chosen due to the skew in the participant scores (Figure 10 *Left*). The data rejects the null hypothesis that the distribution of scores under each problem type are equal with a $p$-value of 0.041.

In addition to the one-sided Mann–Whitney U test, we employed a linear regression with heteroskedasticity-consistent standard errors. This accounted for the fluctuations in problem fre-

| Problem | Human | | | LLM | | | |
|---|---|---|---|---|---|---|---|
| | Original | Obfuscated | Delta | Original | Obfuscated | Delta | Random |
| Warlpiri | 39.88 (n=14) | 44.64 (n=14) | +4.76 | 46.67 | 27.66 | -19.01 | 20.83 |
| Umbrian | 18.67 (n=15) | 11.54 (n=13) | -7.13 | 30.00 | 18.67 | -11.33 | 0.00 |
| Kabyle | 27.69 (n=13) | 12.00 (n=15) | -15.69 | 24.00 | 19.62 | -4.38 | 5.00 |
| Tariana | 7.14 (n=14) | 0.00 (n=14) | -7.14 | 14.07 | 4.13 | -9.95 | 0.00 |
| Ligurian | 22.86 (n=15) | 24.76 (n=15) | +1.90 | 44.76 | 16.94 | -27.82 | 28.57 |
| Amele | 15.00 (n=15) | 3.33 (n=15) | -11.67 | 7.50 | 2.98 | -4.52 | 0.00 |
| All | 21.70 | 16.00 | -5.70 | 27.83 | 15.00 | -12.84 | 9.19 |

Table 13: **Human and LLM performance.** Scores represent the mean score. Sample size is shown in brackets for human performance. LLM scores are calculated as the mean score over all LLMs in the study and all permutations. Random represents the expected score from random answers, given the basic formatting instructions in each question are followed.

| | Model 1 | (SE) | Model 2 | (SE) | Model 3 | (SE) |
|---|---|---|---|---|---|---|
| Intercept | 12.065*** | (3.59) | 12.765*** | (3.81) | 22.723*** | (5.63) |
| Obfuscation | −5.796* | (3.07) | −5.805* | (3.27) | −5.100 | (3.94) |
| Kabyle | 10.326* | (5.50) | 12.395** | (6.09) | 12.939 | (8.17) |
| Ligurian | 14.643*** | (5.22) | 17.276*** | (5.85) | 25.745*** | (6.40) |
| Tariana | −5.595 | (3.46) | −5.938 | (3.88) | 10.610 | (7.85) |
| Umbrian | 5.983 | (4.46) | 8.018 | (5.38) | 4.971 | (6.56) |
| Warlpiri | 33.095*** | (5.27) | 35.553*** | (5.15) | 29.213*** | (6.35) |
| Observations | 172 | | 146 | | 86 | |
| $R^2$ | 0.292 | | 0.326 | | 0.275 | |
| Residual Std. Error | 20.010 | (df=165) | 20.249 | (df=139) | 19.017 | (df=79) |
| F Statistic | 14.126*** | (df=6; 165) | 16.926*** | (df=6; 139) | 7.239*** | (df=6; 79) |

Table 14: **Linear regressions for participant score under different inclusion criteria.** The dependent variable is participant score in all models. The participant inclusion criteria are as follows. Model 1: All participants (n=172). Model 2: Excluding participants who we suspect may have cheated, those who returned more than $50\%$ missing data, and those with mean ChrF score below 10 (n=146). Model 3: Excluding participants who scored below random guessing (n=86). SE: Heteroskedasticity-consistent standard errors. df: Degrees of freedom. $^*p < .1$; $^{**}p < .05$; $^{***}p < .01$

quency in the human data. This analysis provides consistent results, suggesting obfuscation is associated with a 5.80 drop in score ($p$-value of 0.059) (Table 14).

We also examine the performance of LLMs on the same problems, averaging their scores over all permutations. Models average $27.83\%$ and $15.00\%$ on unobfuscated and obfuscated problems, respectively, yielding a mean drop of 12.84 percentage points. This is approximately double the decline found in the human group, supporting the claim that models benefit from prior language exposure when completing the unobfuscated problems.

**Accounting for Suspected Cheating** Since participants took our survey remotely, they had the possibility of cheating. To account for this, our survey contained a trap question designed to establish whether participants had discovered the solutions online or used an LLM. Responses to the trap questions were first automatically scored based on whether they correctly identified the Problemese language, then were evaluated qualitatively and independently by two of this paper's authors. The authors identified 15 and 17 responses, 14 of which were common, and, after discussion, it was decided that all 18 responses should be removed to carry out robustness checks. In addition, participants who had more than $50\%$ missing data, and those with a mean chrF score below 10 (visibly irrelevant answers, e.g. inputting random letters) were also removed. Table 14, Model 2 shows the results excluding these participant groups. The effect on the point estimate and standard error of obfuscation is negligible.

|  | Model 4 | (SE) |
|---|---|---|
| Intercept | 0.068 | (0.40) |
| Obfuscation | −0.689** | (0.34) |
| Kabyle | 0.597 | (0.52) |
| Ligurian | 0.139 | (0.53) |
| Tariana | −1.890*** | (0.70) |
| Umbrian | 0.699 | (0.54) |
| Warlpiri | 1.840*** | (0.63) |
| Observations | 172 | |
| Pseudo $R^2$ | 0.163 | |

Table 15: **A logistic regression predicting above random guessing score.** The dependent variable is a binary variable indicating whether a participant's score was above random. Obfuscated problems are more likely to lead to below random guessing performance with a coefficient significant at the $5\%$ level. SE: Heteroskedasticity-consistent standard errors. df: Degrees of freedom. $^*p < .1$; $^{**}p < .05$; $^{***}p < .01$

**Effect when Only Considering those Scoring Above Random**     We also consider whether the effect of obfuscation remains consistent when we only include participants who scored above random, i.e. who likely understood some of the problem.

First, the likelihood of scoring above random depends on the problem type. In the group taking the unobfuscated problems, $57.0\%$ of participants scored above random. In the group taking the obfuscated problems, only $43\%$ scored above random. Furthermore, a logistic regression predicting above random performance, suggests obfuscation is a significant factor with a $p$-value of $0.045$ (Table 15).

Second, after excluding individuals who scored below random guessing, the data can no longer be treated as the outcomes of an RCT, thus non-stratified tests such as the Mann–Whitney U test are no longer appropriate. A linear regression controlling for differences in problem frequency suggests that the coefficient on obfuscation is no longer significant in this subset ($p$-value of $0.196$), though the point estimate ($−5.100$) remains similar and this result could be an artifact of the small sample size.

**Data Availability**     An anonymised version of the dataset and a notebook to generate results are available in the GitHub repository.

### L.3  EXTERNAL AUDIT OF PROBLEMS

To verify that the problems selected for the RCT remained solvable when rewritten and obfuscated, we asked two International Linguistics Olympiad (IOL) medallists to audit one set of obfuscations each. Both auditors were external to this research project and acted independently. Both had prior familiarity with the original UKLO problems.

The auditors confirmed that all problems remained solvable via the same reasoning steps and were not significantly harder under the new orthographies. One auditor commented that problems appeared to be marginally harder since the languages appeared more 'unfamiliar' and the other commented that they did not believe it made any difference. Both also mentioned that they personally found the problems harder since they knew the original orthographies and thus found the new versions 'disorienting', however this is not an issue that we would expect novice test takers to experience.

### L.4  DISCUSSION

The results suggest that obfuscation causes a small but statistically significant drop in performance when taken by inexperienced test takers. This is despite the required reasoning steps and problem solvability remaining unchanged. The point estimate of the effect is $−5.80$ percentage points: approximately equivalent to half a question per problem. If LLMs behave similarly to participants in this study, then we may expect some performance decrease even in cases where the model did not have prior exposure to the original Problemese language. Therefore, performance drops between unobfuscated and obfuscated problems should not be solely attributed to language exposure. However,

our estimates for the impact of obfuscation are small; thus we would not expect large changes in performance given robust reasoning skills.

We do not have strong evidence for why obfuscation appears to make problems harder. The external audit confirmed that the problems remained solvable via the same reasoning steps, thus any changes to the problem difficulty are superficial only. We speculate that obfuscation may cause the orthography to appear less familiar or naturalistic relative to English, giving the perception that the problems are harder to solve, thus causing participants to exert less effort (Scasserra, 2008). Whilst the mean self-reported difficulty level is marginally higher in the obfuscated group (7.39 versus 7.48, Figure 10), the distributions are not significantly different under a Mann–Whitney U test. However, we do find that participants solving obfuscated problems spent less time on the problem before submitting their answers ($p = 0.071$ under a one-sided Mann–Whitey U test), which would support this hypothesis.

Alternatively, knowledge of English alone may have been sufficient to score marginally higher in the unobfuscated versions of some problems. In particular, Ligurian and Umbrian are both European languages descending from Latin. In Ligurian, this would have increased the familiarity of the language, e.g. *vió:vet:a, 'violet'* or *pásta, 'pasta'*, but would have offered no assistance when solving the reasoning problem. In Umbrian, knowledge of English may have directly helped, especially as the Latin translation were also provided (under the label of 'Language Y'). For example, one question asked for a translation of the Language Y (Latin) words *populum* and *urbs* into both Language X (Umbrian) and English. Knowledge of the English words *population* and *urban* may have helped the participants locate the correct translations, *community* and *town*, from the text provided. While this may have offered marginal assistance in unobfuscated problems, the overall results are largely unchanged when Umbrian is excluded.

**Limitations**    These results are subject to several limitations. First, participants generally found these problems extremely challenging with only $50\%$ of participants scoring above random. The significance of the results is partially driven by differences in scoring below random and it is unclear whether they would hold under different distributions of scores. As with all human evaluations, the results are highly specific to our demographic of participants (inexperienced test takers) and may not generalise outside of this.

Second, we specifically chose relatively easy problems so that participants had a realistic chance of solving them. As a result, these results may not be indicative of the effect of obfuscation across the full distribution of problems.

Third, despite our best efforts to prevent and identify cheating, there is a risk these results are driven by the availability of online materials. This is a problem with any form of online and unproctored assessment, which future studies might wish to address through in-person test taking.

## L.5    INFORMED CONSENT

All participants were required to consent to participation in the study by signing the form below.

---

**Informed consent**

You are being invited to take part in a research project led by the University of Oxford. Before you decide whether to participate, it is important for you to understand why the research is being done and what it will involve. The University of Oxford supports the practice of protecting human participants in research. Please take time to read the following information carefully and discuss it with others if you wish. We appreciate your interest in participating in this online task.

**What is this study?**

This research aims to measure how well people can solve linguistic reasoning problems. A linguistic reasoning problem is a set of questions that requires problem-takers to study a new language that they have likely never seen before. The aim of the problem is to use example translations between English and the second language to decipher how this language works. You will then have to apply your deciphered knowledge to translate new words and phrases. In this research study, the second language may be written in a non-standard way.

---

At the start of the survey, you will be provided with further details about the specific problem. These details contain all information required to solve the task and no background knowledge of languages other than English is required.

When you start the survey, you will be given some basic instructions, then will click through to the main problem. You will have up to 45:00 minutes to solve the question and write down your answers. There is a countdown on the page which shows time remaining. After completing the problem, there is a series of follow-up questions about your experience that will take around 3 minutes to complete. We will collect your responses to the reasoning problem and follow-up questions.

**How will I be compensated?**

Compensation for this task is £7.63. In addition to the base compensation, this study has a bonus system based on performance. Scoring $50\%$ or higher will result in a £2.00 bonus (£9.63 total wage). Scoring $100\%$ will result in a £6.00 bonus (£13.63 total wage). Please note that bonus payments will not be made automatically and may take up to a week to reach your Prolific account.

**Further information and FAQs.**

You have been invited to participate as you are over the age of 18. Please read through this information before agreeing to participate (if you wish to).

You may ask any questions before deciding to take part by contacting the researcher (details below). The research contact is Harry Mayne (harry.mayne@oii.ox.ac.uk), who is a DPhil student at the Oxford Internet Institute at the University of Oxford. This project is being completed under the supervision of Dr Adam Mahdi.

**Do I have to take part?**

No. Please note that participation is voluntary. If you do decide to take part, you may withdraw at any point for any reason before submitting your answers by pressing the 'Exit' button or closing the browser.

**How will my data be used?**

We will not collect any data that could directly identify you. Your IP address will not be stored. We will take all reasonable measures to ensure that data remains confidential.

The responses you provide will be stored in a password-protected electronic file on the University of Oxford's secure servers and may be used in academic publications, conference presentations, reports or websites. Research data with Prolific IDs will be stored internally for three years after publication or public release of the work of the research. De-identified research data, without Prolific IDs, may be publicly released and therefore in the public domain.

The data that we collect from you may be transferred to, stored and/ or processed at a destination outside the UK and the European Economic Area. By submitting your personal data, you agree to this transfer, storing or processing. The results may be written up in partial fulfilment of the requirements for a DPhil degree.

**Who will have access to my data?**

The University of Oxford is the data controller with respect to your personal data and, as such, will determine how your personal data is used in the study. The University will process your personal data for the purpose of the research outlined above. Research is a task that we perform

in the public interest. Further information about your rights with respect to your personal data is available from https://compliance.admin.ox.ac.uk/individual-rights.

The data you provide may be shared with the researchers on this project, Harry Mayne, Dr Adam Mahdi and any other author involved in the publication of the research.

We would also like your permission to use the data in future studies and to share data with other researchers (e.g. in online databases). Data will be de-identified (Prolific IDs removed) before it is shared with other researchers or results are made public.

**Can I withdraw my data?**

Yes, your data can be withdrawn up until 11:59 PM, 1st May 2025. To withdraw your data please contact Harry Mayne (harry.mayne@oii.ox.ac.uk) or Prolific. Your participation in this study is entirely voluntary, and you have the right to withdraw at any time before the deadline without penalty or negative consequences. If you choose to withdraw, any data collected from you will be deleted and not included in the final analysis.

**Who has reviewed this study?**

This project has been reviewed by, and received ethics clearance through a subcommittee of the University of Oxford Central University Research Ethics Committee [Application 1037300].

**Who do I contact if I have a concern or wish to complain?**

If you have a concern about any aspect of this study, please speak to Harry Mayne (harry.mayne@oii.ox.ac.uk) or his supervisor, Dr Adam Mahdi (adam.mahdi@oii.ox.ac.uk) and we will do our best to answer your query. We will acknowledge your concern within 10 working days and give you an indication of how it will be dealt with. If you remain unhappy or wish to make a formal complaint, please contact the Chair of the Research Ethics Committee at the University of Oxford who will seek to resolve the matter as soon as possible:

Social Sciences & Humanities Interdivisional Research Ethics Committee; Email: ethics@socsci.ox.ac.uk; Address: Research Services, University of Oxford, Boundary Brook House, Churchill Drive, Headington, Oxford OX3 7GB

Please note that you may only participate in this survey if you are 18 years of age or older.

I confirm that:

- I have had the opportunity to ask questions and receive satisfactory answers.
- I understand participation in this study is voluntary.
- I understand that I can withdraw my data from the study before 11:59 PM, 1st May 2025 without giving a reason or negative consequences.
- I understand who will have access to personal data provided, how the data will be stored and what will happen to the data at the end of the project.
- I understand how to raise a concern or make a complaint.
- I have read and understood the information on this sheet.

☐ I consent
☐ I do not consent

L.6    PARTICIPANT INSTRUCTIONS AND COMPENSATION

**General Instructions**

Welcome to this reasoning challenge.

When you click forward to the next page, you will be presented with a lingiustic reasoning problem. The problem concerns some unknown language called "Language X", which might be a language with very few or no remaining speakers, and may be written in a non-standard way.

You will receive background information on Language X, along with example English translations. Your task is to decipher how the language works and translate new examples. You are likely to need a pen and paper to work this out.

You have up to 45 minutes to complete this task. Your remaining time will be displayed at the top of the page.

You must spend at least 30 minutes on the task; you will not be able to proceed until this time has elapsed.

Please only submit your answers when you are confident, as you will not be able to go back.

Good luck!

Figure 11: **Participant general instructions.** All participants who consented to the experiment were provided with these instructions.

**Bonus Structure**

Compensation for this task is based on the following bonus structure, where 'Score' is your final score in the reasoning problem.

| Score | Bonus | Total Wage |
|-------|-------|------------|
| 0%+ | – | £7.63 |
| 50%+ | £2.00 | £9.63 |
| 100% | £6.00 | £13.63 |

*As mentioned in the consent form, bonus payments will not be made automatically and may*

*take up to a week to reach your Prolific account.*

Figure 12: **Participant compensation.** Compensation was based on a bonus structure to incentivise effort. Participants were guaranteed at least the UK minimum wage.

# M   DATASET METADATA

Below are details about the benchmark dataset following the `HuggingFace` template.

```
license: cc-by-nc-nd-4.0
task_categories:
 - question-answering
tags:
 - reasoning
 - linguistics
 - benchmark
pretty_name: L2
size_categories:
 - 1K<n<10K
source_datasets:
 - https://huggingface.co/datasets/ambean/lingOly
configs:
 - config_name: default
   data_files:
     - split: test
       path: test_small.zip
extra_gated_prompt: >-
 ### LingOly-TOO LICENSE AGREEMENT

 The LingOly-TOO dataset is distributed under a CC-BY-NC-ND 4.0 license.

 All questions in the LingOly-TOO dataset have been used with the
     permission of
 the original authors. The original authors and the United Kingdom
     Linguistics
 Olympiad may retain rights to control the use, and users of this
     dataset will
 assume liability if they use the dataset beyond the terms of use as
     indicated
 by the benchmark.

 The authors do not take responsibility for any licenses that change
     with time.

 In addition to this license, we ask that uses of the dataset are in
     line with
 the Acceptable Use policy described below.

 ### Acceptable Use Policy

 This dataset is exclusively intended as a benchmark for evaluating
     language
 models subject to the terms of the license. For the integrity of the
 benchmark, users should not:
     * Re-distribute the questions or answers of the benchmark in formats
     (such as plain text) which leak the benchmark to web-scraping.
     * Train language models directly using the content of this benchmark.
extra_gated_fields:
 By clicking Submit below I accept the terms of the license and
     Acceptable Use policy: checkbox
extra_gated_button_content: Submit
```

