# OpenReview forum: "LINGOLY-TOO: Disentangling Reasoning from Knowledge with Templatised Orthographic Obfuscation"
_ICLR.cc/2026/Conference — ICLR 2026 Poster_

### Official Review · Reviewer_afLH · 2025-10-31

**Soundness:** 3
**Presentation:** 3
**Contribution:** 2
**Rating:** 6
**Confidence:** 3

**Summary:**

The paper introduces LINGOLY-TOO, a 6,995-QA benchmark built by applying expert, grapheme-level orthographic obfuscations to 82 UKLO problems, aiming to disentangle reasoning from knowledge/memorization. The authors define clear metrics (Mog vs. Mobf, plus robust variants), run broad model evaluations, and provide validation via auditors and a human RCT. Results show sizable drops under obfuscation (e.g., top model ≈0.60→0.48), correlations with language resourcedness, and modest gains from guided reasoning.

**Strengths:**

* Clear problem framing: measuring reasoning when shortcuts (knowledge, contamination) are minimized is timely and important.
* Methodological originality: the reasoning-equivariant, linguistically-informed permutations are thoughtful and non-trivial; the Turkish vowel harmony example nicely motivates rule design.
* Strong experimental design: multiple families of models, bootstrap analysis, no-context control, tokenization controls, and human study make the case persuasive.

**Weaknesses:**

1. **Insufficient Failure-Mode Analysis**

   The paper documents performance declines but does not deeply probe *why* models fail on obfuscated problems (e.g., difficulty inferring morphemic patterns, inconsistent multi-step reasoning, or fallback to guessing). Please add a qualitative analysis of model outputs—contrasting common errors on obfuscated vs. original items—to connect performance gaps to specific reasoning deficits. Building on this, apply statistical tools (e.g., clustering) to categorize and quantify linguistic reasoning failures.

2. **Limited Accessibility of the Permutation Ruleset**

   The permutation rules (Appendix B) are dense and lack a high-level summary in the main text. A concise overview—such as a table of key constraints and invariances, or a concept diagram illustrating how reasoning equivariance is preserved—would make the method more accessible, especially to readers without a linguistics background.

**Questions:**

1. **On Failure Attribution (related to Weakness 1)**

   Can the authors extend the analysis to *quantify* the causes of reasoning failure? For instance, by labeling dominant error types and reporting their prevalence across models and difficulty levels.

2. **On Ruleset Comprehensibility (related to Weakness 2)**

   While we appreciate the authors’ effort on the permutation design, could the paper include auxiliary tables or diagrams that summarize the rules and constraints at a glance, to help non-linguist researchers quickly understand how reasoning equivariance is enforced?

---

> ### Author Response · Authors · 2025-11-22
>
> Thank you for raising these valuable points.
>
> Understanding the failure modes of reasoning is interesting complementary work to designing LingOly-TOO, however extensive systematic analyses are reserved for future work. Given the timescale it would involve to manually process the outputs and reasoning traces across all problems and models, this was not possible to do at scale as an add-on to our existing work. An attempt to use LLMs to assist in the analysis of reasoning traces was also unsuccessful, since the LLMs were unable to identify erroneous reasoning steps, making both type I and type II errors.
>
> Some limited manual inspection suggested two main failure types. Firstly, the reasoning traces often contain incorrect assertions which are based on one data point (often misinterpreted). These assertions are then treated as true through the rest of the reasoning. This mimics the human pattern of “observe and intuit”, but without checking the conclusions against the data. Secondly, even when exceptions are found elsewhere in the context, the models often ignored these and continued to take the assertions as true. This mostly leads to incorrect final conclusions.
>
> Heuristically, the models’ approach to reasoning appears to rely heavily on confident guess-and-check techniques, rather than the slower more methodical approach required for harder UKLO puzzles.
>
> Regarding Ruleset Comprehensibility, we are glad the reviewer like the Turkish vowel harmony example and agree with the reviewer that it would nice to have a set of rules and constraints that can be inferred “at a glance”. However, there are two reasons why we feel this may not be feasible. First, when designing the rulesets, our experts chose these to permit the maximum number of permutations possible to maintain solvability. Therefore we did not start with a list of heuristics we want to keep, which is what we believe the reviewer is alluding to. Our Turkish vowel harmony is designed to give readers a flavour of this process. Secondly, even where there are linguistic phenomena which guide the the permutation rulesets in multiple problems (such as vowel harmony), they surface very differently in each case. This is due to the unique phonology of each language and the unique structure of each problem. Hence it is not possible to give general constraints which result from even a single linguistic phenomenon.
>
> We do recognise that additional details would be of interest to readers, and propose to write a more reader-friendly introduction to Appendix B and clearer examples in Appendix B.5.

---

### Official Review · Reviewer_cqTp · 2025-10-31

**Soundness:** 3
**Presentation:** 3
**Contribution:** 3
**Rating:** 4
**Confidence:** 3

**Summary:**

This paper introduces LingOLY-TOO, a new benchmark for evaluating reasoning abilities in LLMs. It is built from Linguistics Olympiad problems. The key innovation is the use of "reasoning-equivariant permutations" to obfuscate the problem text. This process changes the orthography but is carefully designed to keep the underlying logical structure and solution steps unchanged. The authors show that while models perform reasonably well on the original problems, their performance drops significantly on the obfuscated versions. This suggests that models rely on prior knowledge and memorization rather than pure reasoning on the original tasks. The paper also includes detailed analysis on the effect of language resourcefulness and a human study.

**Strengths:**

The core idea of the paper is excellent. Using orthographic obfuscation to create a "knowledge-free" test for reasoning is a very clever and direct way to tackle the problem of data contamination. This is a timely and important contribution.

The process for creating the permutations is very rigorous. I am impressed by the careful, manual design of the rulesets by experts to ensure the problems remain solvable. The validation by IOL medallists adds strong credibility to the method.

The experimental section is very comprehensive. The authors evaluated a wide range of models, including the latest reasoning-specific ones. The analysis goes beyond just overall scores to include "no-context" tests, robustness checks, and the correlation with language resources. The human study is also a valuable addition.

**Weaknesses:**

The use of exact match is simple but might be too strict. Sometimes, a model might have the correct reasoning but make a small mistake in formatting the final answer. Using only exact match could penalize such cases.


The human study shows a small but noticeable performance drop (5.7%) for humans on obfuscated problems. This suggests that the obfuscation itself might add some cognitive load, making the problems slightly harder to parse, even for humans who don't rely on prior knowledge of the language.

The process of creating permutation rulesets relies heavily on manual expert work. This might make it difficult to scale the benchmark to a much larger size or to adapt it quickly to other domains.

**Questions:**

1. Have you considered any evaluation metrics other than exact match (e.g., edit distance or partial credit schemes) that could capture instances where the model's reasoning is mostly correct but the final output has a minor error? What are the potential challenges in implementing such metrics for this benchmark?

2. The human study shows a 5.7% performance drop due to obfuscation. Could you discuss a bit more how we should interpret the model's performance drop in light of this? Specifically, how much of the model's drop might be attributed to the increased difficulty of processing the unfamiliar orthography, versus the removal of knowledge-based shortcuts?

3. The permutation rulesets are designed by experts. Do you think this method could be applied to other domains like mathematical reasoning or code generation? What would be the main challenges in designing "reasoning-equivariant permutations" for those domains?

---

> ### Author Response · Authors · 2025-11-23
>
> Thank you for your review. We are glad that you found the contributions of the paper to be valuable, and we also appreciate your questions and concerns that we can address to further increase the impact of our paper.
>
> We agree that there are challenges involved with using exact match as a metric. At present, we discuss the most relevant limitations of exact match, namely uniformly penalising all incorrect answers (including partially correct ones) and limiting insights into failure modes, in Section 6. Previous works in this area, such as LingOly [1] and Linguini [2], have done more extensive testing with other metrics, but ultimately concluded that exact match was necessary for this use case. Metrics such as BLEU and ROUGE are unsuitable for the very short answers common in our benchmark (note from Table 2 that 14% of our answers are single digits) and chrF (and edit distance) are sensitive to repeating words/characters from the context, which is a common behaviour in the models being tested. We provide a specific example below.
>
> Problem 16 4.1 (3) requires a translation of “your (plural) iguana” to Ulwa. The word for iguana in Ulwa is kahma, which is already given in the context. The correct answer is kah**mana**ma, as your (plural) is -mana-. Claude Opus 4.1 returned kahma**na**, which is the wrong form and placement of the possessive. If we assigned partial credit through e.g. edit distance, the answer from Claude will score highly, even though this is an incorrect answer which completely failed to reason. Therefore, assigning partial credit would unnecessarily inflate the baseline through repeating parts of words in the context.
>
> The 5.7% performance decrease observed in the human study also motivated us to explore the potential causes of models performance drop, which was higher in all models (8.6% average drop) and specifically the reasoning variants (9.9% average drop) for the same problems. We are confident that the solution steps and consistency of obfuscated problems remain intact as validated by both the linguistic Olympiad experts in the team and the independent IOL competitors.
>
> We distinguished between two forms of familiarity with orthography: (a) familiarity due to knowledge of the language and its structure (including phonological and lexical knowledge) and b) familiarity from basic exposure to orthography’s token sequences. We posit (a) to be part of the knowledge that models can use to shortcut reasoning and consider obfuscation a counter measure. We show in the language-resourcedness experiment that such familiarity might indeed offer partial explanation.
>
> In relation to (b), if performance drop can be attributed to failure in processing tokens from unfamiliar orthographies, we would expect to see such drop across languages and problems. We do not see such consistent effect. In several problems and for different models, performance does not drop across all obfuscation versions (see also Variance across permutations, p.8). In other analyses we carried out, we observed that altering the tokenisation (Table 3/Appendix K) shows no improvement but providing expert guidance steps derived from unobfuscated versions seems to help models reason on obfuscated versions.
>
> Recognisability of a linguistic reasoning task (e.g. translation) can render the task trivial to models with prior knowledge of the languages. Compared to other domains such as mathematics, it is easier to guarantee a model has no prior knowledge of the perturbed version of a language. This distinction helps expose model tendency to rely on prior knowledge instead of reasoning. For coding or mathematics, we do not anticipate particular difficulties for similar reasoning-equivariant permutation to performed, and there has been a benchmark developed in the last few months involving permuting variable names and values [3]. However, we do not expect similar guarantees of unrecognisability with obfuscation as in LingOly-TOO. Indeed, the drops seen in [3] is in general smaller than the gap seen in LingOly-TOO.
>
> [1] Bean, Andrew M., et al. "LINGOLY: A benchmark of olympiad-level linguistic reasoning puzzles in low resource and extinct languages." Advances in Neural Information Processing Systems 37 (2024): 26224-26237.
>
> [2] Sánchez, Eduardo, et al. "Linguini: A benchmark for language-agnostic linguistic reasoning." *arXiv preprint arXiv:2409.12126* (2024).
>
> [3] Hao, Yuren, Xiang Wan, and Chengxiang Zhai. "An Investigation of
> Robustness of LLMs in Mathematical Reasoning: Benchmarking with
> Mathematically-Equivalent Transformation of Advanced Mathematical
> Problems." *arXiv preprint arXiv:2508.08833* (2025).

---

### Official Review · Reviewer_SdFn · 2025-10-31

**Soundness:** 3
**Presentation:** 3
**Contribution:** 2
**Rating:** 6
**Confidence:** 3

**Summary:**

This paper introduces a challenging reasoning benchmark of 7k question answer pairs, built by applying grammeme-level obfuscations to Linguistic Olympiad problems.  The motivation for building a new dataset is that models rely on prior language knowledge learnt via pre-training. This dataset is built to test out models' reasoning capabilities and removes any cues that could trigger memorized translations.

**Strengths:**

1. Release of a large-sized (7k) benchmark dataset that clearly separates the reasoning abilities from memorized knowledge.
2. In-depth analysis: The authors conducted various analyses, such as the ability of the model to reason, the effect of tokenization on uncommon characters, and the various across different permutations.
3. Release of dataset and code

**Weaknesses:**

1. In the no-context setting, the difference between the original and the obfuscated dataset is very less; how does the author come to a concrete conclusion?
2. To check the effect of performance drop with uncommon characters, why don't we replace the context with random but real tokens that are not part of the training set? Similar to the ProntoQA dataset, where entities are replaced with false ontology.

**Questions:**

The authors manually created rulesets for this dataset, which makes it harder to extend this work to other datasets. Did the authors try to use LLMs as annotators to see how feasible it is to extend them to other domains/ datasets?

---

> ### Author Response · Authors · 2025-11-20
>
> Thank you for your review. Please find responses to your concerns below.
>
> **No Context**
>
> The results for the no context analysis appear in Table 2. In this table, we see that across all questions, GPT-4o falls from $0.08$ to $0.02$ after obfuscation. In the No Context setting, the expected score is close to zero (and essentially is zero for “Other” answer type where there are no multiple choice options given). The score of $0.08$ is therefore a strong indication that the model has internal knowledge which is contributing to overall performance. By contrast, the score of $0.02$ is better in line with expectations for random guessing. With a standard error $<0.01$, the difference between these two scores is highly significant, allowing us to confidently conclude that the performance difference is not random. For Llama 3.3, the performance on original problems is lower and so the effect size is smaller, but the difference between the original and obfuscated versions is still greater than two standard errors apart.
>
> **Dataset Creation**
>
> We appreciate both of your suggestions here (in Weaknesses and Questions), and found them interesting.
>
> The comparison with PrOntoQA is a good example of an earlier work using structured templates to create puzzles, and we will add it to our related works. We would like some more clarification of your suggestions. Please can you expand on “why don't we replace the context with random but real tokens that are not part of the training set”. Is the idea to obfuscate randomly, rather than with the curated rulesets?
>
> With regards to using LLMs as annotators to extend the task, we are sceptical of the feasibility of this approach with current models. Creation of a permutation ruleset requires both knowledge of the solution to the problem (sometimes including information not found in the official UKLO commentary of the problem) as well as a detailed understanding of the structure and various solution paths of the problem. We find it unlikely that a model which cannot reliably solve the problems could create functioning rulesets. A recent work experimented with using LLMs to generate (easy) Linguistic Olympiad problems, and they largely failed [1].
>
> We believe this expert knowledge that was required to create the rulesets for the benchmark is a strength, and point out that previous labelled datasets and benchmarks like humanity’s last exam [2] and GPQA [3] also require subject matter expert and cannot be arbitrarily extended (the latter hiring 61 contractors with PhDs to write and validate the dataset), but still have high utility and impact. That being said, in areas with less specialised knowledge, such as school-level mathematics, using LLMs to create templates does sound like a promising direction for future work that we will note in our discussion.
>
> [1] Majmudar, Neh and Filatova, Elena. Can LLMs Generate and Solve Linguistic Olympiad Puzzles?. In Proceedings of the 2025 Conference on Empirical Methods in Natural Language Processing, page s 19174–19211, Suzhou, China. Association for Computational Linguistics. (2025) https://aclanthology.org/2025.emnlp-main.969/
>
> [2] Phan, Long, et al. "Humanity's last exam." *arXiv preprint arXiv:2501.14249* (2025).
>
> [3] Rein, David, et al. "GPQA: A Graduate-level Google-proof Q&A Benchmark." *First Conference on Language Modeling*. (2024).

---

### Official Review · Reviewer_eZVp · 2025-11-01

**Soundness:** 2
**Presentation:** 3
**Contribution:** 2
**Rating:** 4
**Confidence:** 5

**Summary:**

This paper introduces LingOly-TOO, a challenging reasoning benchmark that obfuscate Linguistics Olympiad Problems to avoid advantaging models that are using shortcuts such as memorisation and knowledge. The obfuscations preserve the underlying solution logic while removing orthographic clues that could trigger patterns from memorisation or knowledge. Without surprise, the performance of models drastically decrease.

The authors defined knowledge as information stored in model parameters after training, which captures linguistic, factual, and commonsense patterns useful for downstream tasks, and memorisation as when models exploit contaminated datasets, reporting answers previously seen in training

The authors adapted 82 problems from the UKLO, and obfuscate the problem to avoid models relying on linguistic patterns. More specifically, the authors manually created a ruleset for each problem to generate valid permutations of targeted tokens. They apply extra-care to keep names of people, sacred places, etc intact. Overall, the authors generate 6 valid permutations per problem and generate obfuscated versions by altering the problem text. Overall, there are 6995 question-answer pairs.

The authors propose two metrics: the average exact match score across all questions in all permutations and the average exact match across all questions in the unpermuted problem.

In terms of experiments, the authors evaluate 15 reasoning models. The performance varies between the original and obfuscated variants, with GPT and Claude being the most robust models. The authors do a detailed analysis to measure the gap between reasoning and knowledge. It would be great to add a case study to support the findings. In terms of metrics, it would be good to compare human performance on the task and include more details (e.g., how do they compare with LLMs) - I'm aware of Appendix L but would like to see more. Moreover, it would be fairer to assess the performance of models with pass@k.

Overall, I am lukewarm about this paper. Creating obfuscating variants of problems does not seem to relate to real-life tasks, even for measuring reasoning. I have the feeling that this benchmark is focusing on a unrealistic problem / not a real problem. I appreciate the analysis and experiments of the authors, and the paper is very well written and structured.

**Strengths:**

- Various models are used in the benchmark.
- Detailed analysis.

**Weaknesses:**

- I have the feeling that this benchmark focuses on an unrealistic problem / does not represent real life tasks. We cannot expect LLMs to necessarily perform good on those.
- Please add human evaluation + pass@k metrics

**Questions:**

See above.

---

> ### Author Response · Authors · 2025-11-20
>
> Thank you for your detailed summary and review.
>
> With regard to the nature of the task, we believe that logic reasoning capabilities are important to the development of AI, and that axiomatic logic should be generalisable and context independent. To this end, benchmarks like LingOly-TOO are essential in order to test whether training has been successful to teach logical reasoning, or just memorise patterns. In practice, Linguistic Olympiad puzzles are already being used at major ML/NLP venues, including the machine translation shared task at EMNLP [1]. This task is based on similar problems from precursors of LingOly-TOO (namely LingOly [2] and Linguini [3]) for machine translation, linguistic reasoning, open-ended generation, cross-lingual summarization, and LLM-as-a-judge,  and highlighted the limitations of automatic evaluation and persistent challenges in reasoning. LingOly-TOO is specifically valuable since it consists of tasks aimed at high-schoolers with no expert knowledge, and therefore it is informative that models perform worse on it than other benchmarks that would have higher requirements of human subjects.
>
> On the human evaluations, we would be curious to know what additional information you are looking for. As you note, we already have a human baseline in Appendix L. Please can you provide more guidance as to what you mean by “I would like to see more”?
>
> Finally, for pass@k metrics, we have practical considerations for not using this metric, but are interested in your thoughts. First of all, a concern with pass@k is that many of our puzzles involve relatively short/multiple choice answers, therefore it is more likely that a model reaches a correct answer by random chance than would be expected in a longer form generation like coding. For example, for a question with $4$ choices, a random guesser has $0.25$ chance of getting it correctly. On this random guesser, the true pass@k metric, for $k=3$, would be $0.58 \gg 0.25$. Additionally, the models already achieve moderate scores, so we are not convinced about the benefits of a more forgiving metric relative to the issues it raises. Finally, pass@k is obviously much more expensive than pass@1. The cost of running these models on 7k question items is prohibitive, particularly for the thinking models. Could you expand on the benefit that you would expect to gain from adding in pass@k?
>
> [1] Kocmi et al. (2025) Findings of the WMT25 Multilingual Instruction Shared Task: Persistent Hurdles in Reasoning, Generation, and Evaluation. Proceedings of the Tenth Conference on Machine Translation (WMT), Volume 2: Shared Task Papers, pages 462–483. https://aclanthology.org/anthology-files/pdf/wmt/2025.wmt-1.23.pdf
>
> [2] Bean, Andrew M., et al. "LINGOLY: A benchmark of olympiad-level linguistic reasoning puzzles in low resource and extinct languages." Advances in Neural Information Processing Systems 37 (2024): 26224-26237.
>
> [3] Sánchez, Eduardo, et al. "Linguini: A benchmark for language-agnostic linguistic reasoning." *arXiv preprint arXiv:2409.12126* (2024).

---

### Author Response · Authors · 2025-12-03

# Overview

We introduce LingOly-TOO, a benchmark using reasoning-equivariant orthographic obfuscations of Linguistics Olympiad problems to disentangle reasoning from knowledge/memorisation in LLMs.

Reviewers cqTp and afLH describe our orthographic obfuscation as "very clever", "a timely and important contribution", and “methodologically original”. Reviewers also characterise our benchmark construction and experiments as "in-depth" (SdFn), "strong" (afLH, cqTp), "rigorous" and "comprehensive" (cqTp).

We summarise the main concerns and responses, showing why LingOly-TOO is a valuable contribution for measuring reasoning in LLMs.

# Practical Relevance

Reviewer eZVp worried that the benchmark is not representative of real-life tasks. Axiomatic logical reasoning should be context-agnostic, and linguistics puzzles are used at major ML/NLP venues to test model abilities. LingOly-TOO is designed as a controlled diagnostic of reasoning under minimal contamination of knowledge shortcuts. Our finding that LLMs underperform on tasks solvable by high-school students with no linguistic expertise is itself informative about current reasoning capabilities.

# Metric Choices

While we agree with reviewer cqTp that exact match is strict, many answers in our benchmark are short, so alternative metrics like BLEU, ROUGE, or edit distance are unsuitable. Models frequently repeat substrings from the context without reasoning: our Ulwa example shows that partial credit would inflate baselines, whereas exact match is the clearest signal that the model completed the reasoning chain.

Reviewer eZVp suggested evaluating with pass@k, but:
1. many questions have very short answers, so with random guessing pass@k approaches 1 for modest k;
2. models already achieve moderate scores, a more forgiving metric would inflate scores without improving interpretability;
3. computing pass@k for ~7k questions across all models is prohibitively costly

# Interpreting Performance in No-Context Study

Reviewer SdFn questioned how the small performance gap between original and obfuscated problems can support a strong claim about knowledge vs reasoning. Although the expected score should be random chance, GPT-4o achieved many standard deviations above chance on the original problems compared to its performance after obfuscation, strongly suggesting that it is drawing on memorised or parametric knowledge.

# Interpreting Performance with the human Study

Reviewer cqTp asked how much of the performance drop comes from parsing unfamiliar orthography versus removing knowledge shortcuts. We distinguish two kinds of “familiarity”:
a) Linguistic familiarity: knowledge about the language (e.g. morphology) that makes the task easier.
b) Surface/token familiarity: seeing similar token sequences, independent of any linguistic knowledge that aids parsing orthographies.

If the drop were mainly due to b) generic difficulty with unfamiliar tokens, we’d expect consistent drops across permutations and languages, which we do not observe. Altering tokenisation alone does not close the gap, whereas providing expert reasoning steps derived from the original problem helps models on obfuscated versions. LLMs, especially reasoning LLMs, exhibit larger decrease than humans, even though humans also incur additional parsing load.

# Dataset Construction and Scalability
Reviewers SdFn and cqTp remarked manual rulesets may be hard to scale or extend to new domains like maths and asked whether LLMs can help.

LingOly-TOO is a carefully crafted benchmark, not an auto-scalable data factory. We view expert curation as a feature, not a limitation: several impactful benchmarks (e.g. GPQA, “Humanity’s Last Exam”) require substantial expert effort, are not trivially extensible, yet are widely used as high-value stress tests rather than mass-scale datasets. For other domains, similar obfuscations are feasible but we do not expect strong guarantee of "unrecognisability" as in obfuscated orthographies. Preliminary attempts to use current LLMs for ruleset construction were unsuccessful.

Reviewer afLH complimented our vowel harmony example but noted that Appendix B is dense and would benefit from a higher-level summary. However, different languages and problems impose different phonological and structural constraints, so a single concise global schema cannot be provided. We will add a clearer introduction to Appendix B to explain the general workflow and recurring phenomena and provide clearer examples to show how rulesets preserve reasoning structure.

# Failure-Mode Analysis

Reviewer afLH noted we show performance declines but do not deeply analyse why. We conducted limited manual inspection that suggested two failure modes: (a) models often form a wrong early hypothesis, (b) they ignore counterexamples that appear later in the context. Using LLMs to scale these analyses was not reliable. Given the benchmark’s size, a full analysis is resource-intensive and deferred to future work.

---

### Meta-Review · Area_Chair_1WZ6 · 2026-01-06

**Summary:**

This work introduces a new benchmark designed to assess whether LLMs rely on shortcuts during problem-solving. While reviewers agreed on the soundness of the experimental design, the primary remaining concern centers on the task's ecological validity. Specifically, whether it accurately represents real-world challenges. While this critique holds merit, I believe the community benefits from a diverse ecosystem of benchmarks, provided the underlying methodology is robust.

**Reviewer Concerns:**

Remaining concerns (projected by the AC):

1. There could be more human evaluation.

2. There is no pass@k metrics.

3. This datasets does not represent real life tasks.

**Reviewer Scores:**

6/6/6/4

---

### Decision · Program_Chairs · 2026-01-26

Accept (Poster)